# Denoising Diffusion Error Correction Codes

**Yoni Choukroun**
The Blavatnik School of Computer Science
Tel Aviv University
`choukroun.yoni@gmail.com`

**Lior Wolf**
The Blavatnik School of Computer Science
Tel Aviv University
`wolf@cs.tau.ac.il`

## Abstract

Error correction code (ECC) is an integral part of the physical communication layer, ensuring reliable data transfer over noisy channels. Recently, neural decoders have demonstrated their advantage over classical decoding techniques. However, recent state-of-the-art neural decoders suffer from high complexity and lack the important iterative scheme characteristic of many legacy decoders. In this work, we propose to employ denoising diffusion models for the soft decoding of linear codes at arbitrary block lengths. Our framework models the forward channel corruption as a series of diffusion steps that can be reversed iteratively. Three contributions are made: (i) a diffusion process suitable for the decoding setting is introduced, (ii) the neural diffusion decoder is conditioned on the number of parity errors, which indicates the level of corruption at a given step, (iii) a line search procedure based on the code's syndrome obtains the optimal reverse diffusion step size. The proposed approach demonstrates the power of diffusion models for ECC and is able to achieve state of the art accuracy, outperforming the other neural decoders by sizable margins, even for a single reverse diffusion step. Our code is attached as supplementary material.

## 1 Introduction

Reliable digital communication is of major importance in the modern information age and involves the design of codes that can be robustly decoded despite noisy transmission channels. The target decoding is defined by the NP-hard maximum likelihood rule, and the efficient decoding of commonly employed families of codes, such as algebraic block codes, remains an open problem.

Recently, powerful learning-based techniques have been introduced. Model-free decoders (O'Shea & Hoydis, 2017; Gruber et al., 2017; Kim et al., 2018) employ generic neural networks and may potentially benefit from the application of powerful deep architectures that have emerged in recent years in various fields. A Transformer-based decoder that is able to incorporate the code into the architecture has been recently proposed by Choukroun & Wolf (2022). It outperforms existing methods by sizable margins, at a fraction of their time complexity. The decoder's objective in this model is to predict the noise corruption, to recover the transmitted codeword (Bennatan et al., 2018).

Deep generative neural networks have shown significant progress over the last years. Denoising Diffusion Probabilistic Models (DDPM) (Ho et al., 2020b) are an emerging class of likelihood-based generative models. Such methods use diffusion models and denoising score matching to generate new samples, for example, images (Dhariwal & Nichol, 2021) or speech (Chen et al., 2020a). The DDPM model learns to perform a reversed diffusion process on a Markov chain of latent variables, and generates samples by gradually removing noise from a given signal.

One major drawback of model-free approaches is the high space/memory requirement and time complexity that hamper its deployment on constrained hardware. Moreover, the lack of an iterative solution means that both highly and slightly corrupted codewords go through the same computationally demanding neural decoding procedure.

In this work, we consider the error correcting code paradigm via the prism of diffusion processes. The channel codeword corruption can be viewed as an iterative forward diffusion process to be

reversed via an adapted DDPM. As far as we can ascertain, this is the first adaptation of diffusion models to error correction codes.

Beyond the conceptual novelty, we make three technical contributions: (i) our framework is based on an adapted diffusion process that simulates the coding and transmission processes, (ii) we further condition the denoising model on the number of parity-check errors, as an indicator of the signal's level of corruption, and (iii) we propose a line-search procedure that minimizes the denoised code syndrome, in order to provide an optimal step size for the reverse diffusion.

Applied to a wide variety of codes, our method outperforms the state-of-the-art learning-based solutions by very large margins, employing extremely shallow architectures. Furthermore, we show that even a single reverse diffusion step with a controlled step size can outperform concurrent methods.

## 2 RELATED WORKS

The emergence of deep learning for communication and information theory applications has demonstrated the advantages of neural networks in many tasks, such as channel equalization, modulation, detection, quantization, compression, and decoding (Ibnkahla, 2000). Model-free decoders employ general neural network architectures (Cammerer et al., 2017; Gruber et al., 2017; Kim et al., 2018; Bennatan et al., 2018). However, the exponential number of possible codewords makes the decoding of large codes unfeasible. Bennatan et al. (2018) preprocess the channel output to allow the decoder to remain provably invariant to the transmitted codeword and to eliminate risks of overfitting. Model-free approaches generally make use of multilayer perceptron networks or recurrent neural networks to simulate the iterative process existing in many legacy decoders (Gruber et al., 2017; Kim et al., 2018; Bennatan et al., 2018). However, many architectures have difficulties in learning the code or analyzing the reliability of the output, and require prohibitive parameterization or expensive graph permutation preprocessing (Bennatan et al., 2018).

Recently, Choukroun & Wolf (2022) proposed the Error Correction Code Transformer (ECCT), obtaining SOTA performance. The model embeds the signal elements into a high-dimensional space where analysis is more efficient, while the information about the code is integrated via a masked self-attention mechanism.

Diffusion Probabilistic Models were first introduced by Sohl-Dickstein et al. (2015), who presented the idea of using a slow iterative diffusion process to break the structure of a given distribution while learning the reverse neural diffusion process, in order to restore the structure in the data. Song & Ermon (2019) proposed a new score-based generative model, building on the work of Hyvärinen & Dayan (2005), as a way of modeling a data distribution using its gradients, and then sampling using Langevin dynamics (Welling & Teh, 2011).

The DDPM method of Ho et al. (2020b) is a generative model based on the neural diffusion process that applies score matching for image generation. Song et al. (2020b) leverage techniques from stochastic differential equations to improve the sample quality obtained by score-based models; Song et al. (2020a) and Nichol & Dhariwal (2021a) propose methods for improving sampling speed; Nichol & Dhariwal (2021a) and Saharia et al. (2021) demonstrated promising results on the difficult ImageNet generation task, using upsampling diffusion models. Several extensions to other fields, such as audio (Kong et al., 2020; Chen et al., 2020b), have been proposed.

## 3 BACKGROUND

We provide in this section the necessary background on error correction coding and DDPM.

**Coding** We assume a standard transmission that uses a linear code $C$. The code is defined by the binary generator matrix $G$ of size $k \times n$ and the binary parity check matrix $H$ of size $(n-k) \times n$ defined such that $GH^T = 0$ over the order 2 Galois field $GF(2)$.

The input message $m \in \{0,1\}^k$ is encoded by $G$ to a codeword $x \in C \subset \{0,1\}^n$ satisfying $Hx = 0$ and transmitted via a Binary-Input Symmetric-Output channel, e.g., an AWGN channel. Let $y$ denote the channel output represented as $y = x_s + \varepsilon$, where $x_s$ denotes the Binary Phase Shift Keying (BPSK) modulation of $x$ (i.e., over $\{\pm 1\}$), and $\varepsilon$ is a random noise independent of

the transmitted $x$. The main goal of the decoder $f : \mathbb{R}^n \to \mathbb{R}^n$ is to provide a soft approximation $\hat{x} = f(y)$ of the codeword.

We follow the preprocessing of Bennatan et al. (2018); Choukroun & Wolf (2022), in order to remain provably invariant to the transmitted codeword and to avoid overfitting. The preprocessing transforms $y$ to a vector of dimensionality $2n - k$ defined as

$$\tilde{y} = h(y) = [|y|, s(y)] , \tag{1}$$

where, $[\cdot, \cdot]$ denotes vector concatenation, $|y|$ denotes the absolute value (magnitude) of $y$ and $s(y) \in \{0, 1\}^{n-k}$ denotes the binary code *syndrome*. The syndrome is obtained via the $GF(2)$ multiplication of the binary mapping of $y$ with the parity check matrix such that

$$s(y) = Hy_b := H\mathrm{bin}(y) := H\big(0.5(1 - \mathrm{sign}(y))\big). \tag{2}$$

The induced parameterized decoder $\epsilon_\theta : \mathbb{R}^{2n-k} \to \mathbb{R}^n$ with parameters $\theta$ aims to predict the multiplicative noise denoted as $\tilde{\varepsilon}$ and defined such that $y = x_s \cdot \tilde{\varepsilon}$. The prediction of the multiplicative noise instead of the additive physical one is done in order to remain invariant to the transmitted codeword ($|y| = |x_s \tilde{\varepsilon}| = |\tilde{\varepsilon}|$), thereby avoiding the risk of code overfitting, as described by Bennatan et al. (2018) and the proof of lemma 1 of Richardson & Urbanke (2001). The final prediction takes the form $\hat{x}_s = \mathrm{sign}(y \cdot \epsilon_\theta(|y|, Hy_b))$.

**Denoising Diffusion Probability Model (DDPM)**  Ho et al. (2020a) assume a data distribution $x_0 \sim q(x)$ and a Markovian noising process $q$ that gradually adds noise to the data to produce noisy samples $\{x_i\}_{i=1}^T$. Each step of the corruption process adds Gaussian noise according to some variance schedule given by $\beta_t$ such that

$$
\begin{aligned}
q(x_t|x_{t-1}) &\sim \mathcal{N}(x_t; \sqrt{1 - \beta_t} x_{t-1}, \beta_t I)) \\
x_t &= \sqrt{1 - \beta_t} x_{t-1} + \sqrt{\beta_t} z_{t-1}, \ z_{t-1} \sim \mathcal{N}(0, I).
\end{aligned}
\tag{3}
$$

$q(x_t|x_0)$ can be expressed as a Gaussian distribution such that, with $\alpha_t := 1 - \beta_t$ and $\bar{\alpha}_t := \prod_{s=0}^t \alpha_s$, we have

$$
\begin{aligned}
q(x_t|x_0) &\sim \mathcal{N}(x_t; \sqrt{\bar{\alpha}_t} x_0, (1 - \bar{\alpha}_t)I) \\
x_t &= \sqrt{\bar{\alpha}_t} x_0 + \varepsilon \sqrt{1 - \bar{\alpha}_t}, \ \varepsilon \sim \mathcal{N}(0, I).
\end{aligned}
\tag{4}
$$

The intractable reverse diffusion process $q(x_{t-1}|x_t)$ approaches a diagonal Gaussian distribution as $\beta_t \xrightarrow[t \to \infty]{} 0$ (Sohl-Dickstein et al., 2015) and can be approximated using a neural network $p_\theta(x_t)$ in order to predict the Gaussian statistics. The model is trained by stochastically optimizing the random terms of the variational lower bound of the negative log-likelihood function.

One can find via Bayes' theorem that the posterior $q(x_{t-1}|x_t, x_0)$ is also Gaussian, making the objective a sum of tractable KL divergences between Gaussians. Ho et al. (2020a) found a more practical objective, defined via the training of a model $\epsilon_\theta^{DDPM}(x_t, t)$ that predicts the additive noise $\varepsilon$ from Eq. 4 as follows

$$\mathcal{L}_{DDPM}(\theta) = \mathbb{E}_{t \sim \mathcal{U}[1,T], x_0 \sim q(x), \varepsilon \sim \mathcal{N}(0,I)} ||\varepsilon - \epsilon_\theta^{DDPM}(x_t, t)||^2. \tag{5}$$

The distribution $q(x_T)$ is assumed to be a nearly isotropic Gaussian distribution, such that sampling $x_T$ is trivial. Thus, the reverse diffusion process is given by the following iterative process

$$x_{t-1} = \frac{1}{\sqrt{\alpha_t}} \left( x_t - \frac{1 - \alpha_t}{\sqrt{1 - \bar{\alpha}_t}} \epsilon_\theta^{DDPM}(x_t, t) \right). \tag{6}$$

## 4 DENOISING DIFFUSION ERROR CORRECTION CODES

We present the elements of the proposed denoising diffusion for decoding and the proposed architecture, together with its training procedure. An illustration of the coding setting and the proposed decoding framework are given in Figure 1.

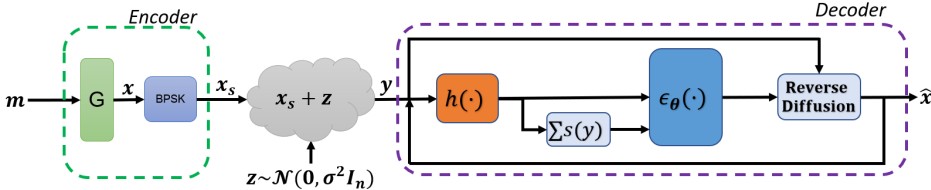

Figure 1: Illustration of the communication system. We train a parameterized iterative decoder $\epsilon_\theta$ conditioned on the number of parity check errors. The decoding is performed iteratively through the reverse diffusion process, as described in this paper.

## 4.1 DATA TRANSMISSION AS A FORWARD DIFFUSION PROCESS

Given a codeword $x_0$ sampled from the Code distribution $x_0 \sim q(x)$, we propose to define the codeword transmission procedure $y = x_0 + \sigma\varepsilon$ as a forward diffusion process adding a small amount of Gaussian noise to the sample in $t$ steps with $t \in (0, \ldots, T)$, where the step sizes are controlled by a variance schedule $\{\beta_t\}_{t=0}^T$. In our setting, we propose the following *unscaled* forward diffusion

$$q(x_t := y|x_{t-1}) \sim \mathcal{N}(x_t; x_{t-1}, \beta_t I). \tag{7}$$

Thus, for a given received word $y$ and a corresponding $t$, we consider $y$ as a codeword that has been corrupted gradually, such that for $\varepsilon \sim \mathcal{N}(0, I)$

$$y := x_t = x_0 + \sigma\varepsilon = x_0 + \sqrt{\bar{\beta}_t}\varepsilon \sim \mathcal{N}(x_t; x_0, \bar{\beta}_t I), \tag{8}$$

where $\bar{\beta}_t = \sum_{i=1}^t \beta_i$ and $\sigma$ defines the level of corruption of the AWGN channel. Thus, the transmission of data over noisy communication channels can be defined as a *modified* iterative diffusion process to be reversed for decoding.

## 4.2 DECODING AS A REVERSE DIFFUSION PROCESS

Following Bayes' theorem, the posterior $q(x_{t-1}|x_t, x_0)$ is a Gaussian such that $q(x_t|x_{t-1}, x_0) \sim \mathcal{N}(x_t; \tilde{\mu}_t(x_t, x_0), \tilde{\beta}_t I)$, where, according to Eq. 8, we have

$$\tilde{\mu}_t(x_t, x_0) = \frac{\bar{\beta}_t}{\bar{\beta}_t + \beta_t} x_t + \frac{\beta_t}{\bar{\beta}_t + \beta_t} x_0 = x_t - \frac{\sqrt{\bar{\beta}_t}\beta_t}{\bar{\beta}_t + \beta_t}\varepsilon, \text{ and } \tilde{\beta}_t = \frac{\bar{\beta}_t \beta_t}{\bar{\beta}_t + \beta_t}. \tag{9}$$

The full derivation is given in the Appendix A. Similarly to (Sohl-Dickstein et al., 2015; Ho et al., 2020b), we wish to approximate the intractable Gaussian reverse diffusion process $q(x_{t-1}|x_t)$ such that

$$q(x_{t-1}|x_t) \approx p_\theta(x_{t-1}|x_t) \sim \mathcal{N}(x_{t-1}; \mu_\theta(x_t, t), \tilde{\beta}_t I), \tag{10}$$

with fixed variance $\tilde{\beta}_t$. Following the simplified objective of Ho et al. (2020b), one would adapt the negative log-likelihood approximation such that the decoder predicts the additive noise of the adapted diffusion process and

$$\mathcal{L}(\theta) = \mathbb{E}_{t \sim \mathcal{U}[1,T], x_0 \sim q(x), \varepsilon \sim \mathcal{N}(0,I)}[||\varepsilon - \epsilon_\theta(x_0 + \sqrt{\bar{\beta}_t}\varepsilon, t)||^2]. \tag{11}$$

One interesting property of the syndrome-based approach of Bennatan et al. (2018) is that, similarly to denoising diffusion models, in order to retrieve the original codeword, the decoder's objective is to predict the channel's noise. However, the syndrome-based approach enforces the prediction of the multiplicative noise $\tilde{\varepsilon}$ instead of the additive noise $\varepsilon$, in contrast to classic diffusion models. We note, however, that the exact value of the multiplicative noise is not important for hard decoding, but only its sign since $x_s = \text{sign}(y\tilde{\varepsilon})$.

Therefore, we propose to learn the hard (i.e., the sign) prediction of the multiplicative noise using the binary cross entropy loss as a surrogate objective, such that

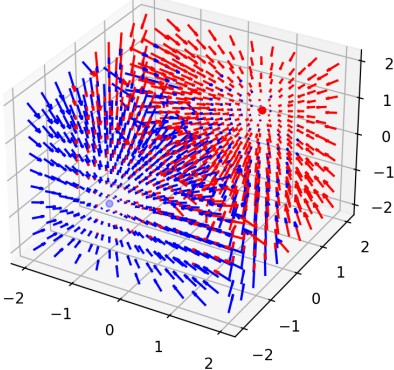

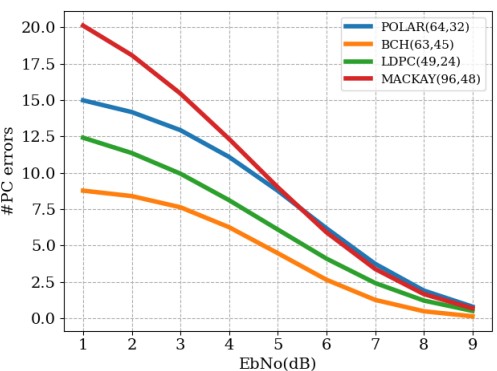

Figure 2: Reverse diffusion dynamics on a (3,1) repetition code. The two points represent the two only signed codewords: $\pm(1,1,1)$. The colors are defined by Maximum Likelihood decoding. Evidently, the denoising diffusion model reverses noisy codes towards the right distribution. An illustration of the forward process for this code is provided in Appendix B.

Figure 3: Influence of the noise or $E_b/N_0$ (normalized SNR) on the number of parity check errors for several codes. The greater the noise, the higher the number of parity check errors, which demonstrates that the syndrome conveys information about the level of noise.

$$\mathcal{L}(\theta) = \mathbb{E}_{t,x_0,\varepsilon} \text{BCE}\Big(\epsilon_\theta(x_0 + \sqrt{\bar{\beta}_t}\varepsilon, t), \tilde{\varepsilon}_b\Big), \tag{12}$$

where the target binary multiplicative noise is defined as $\tilde{\varepsilon}_b = \text{bin}\big(x_0(x_0 + \sqrt{\bar{\beta}_t}\varepsilon)\big)$, and BCE denotes the binary cross entropy loss.

## 4.3 Denoising via Parity Check Conditioning

The reverse denoising process of traditional DDPM is conditioned by the time step. Thus, by sampling Gaussian noise, which is assumed as equivalent to step $t = T$, one can fully reverse the diffusion by up to $T$ iterations. In our case, we are not interested in a generative model, but in an *exact* iterative denoising scheme, where the original signal is only corrupted to a measured extent.

Moreover, a given noisy code conveys information about the level of noise via its syndrome, since $s(y) = Hy = Hx + Hz = Hz$. Fig. 3 illustrates the impact of noise on the number of parity check errors. Evidently, one can *approximate* an injective function between the number of parity check errors and the amount of noise. Such a function is a direction indication of the proximity of the current iterate to a solution (codeword). Therefore, we suggest conditioning the diffusion decoder according to the number of parity check errors $e_t$, such that $e_t := e(x_t) = \sum_{i=1}^{n-k} s(x_t)_i \in \{0, \ldots n-k\}$. The resulting training objective is now given by $\mathcal{L}(\theta) = \mathbb{E}_{t,x_0,\varepsilon} \text{BCE}\Big(\epsilon_\theta(x_0 + \bar{\beta}_t^{1/2}\varepsilon, \boldsymbol{e_t}), \tilde{\varepsilon}_b\Big)$.

Following this logic, the number of required denoising steps $T = n - k$ is set as the maximum number of parity check errors. Similarly to the classical DDPM training procedure, sampling a time step $t \sim \mathcal{U}(0, \ldots, T)$ produces noise, which in turn induces a certain number of parity errors.

The training procedure of our method is given in Alg. 1. The framework assumes a random "time" sampling, producing a noise and then a syndrome to be corrected. Note that, our model-free solution is invariant to the transmitted codeword, and the diffusion decoding can be trained with one single codeword (Alg. 1 line 1).

Since the denoising model predicts the multiplicative noise $\tilde{\varepsilon}$, at inference time it needs to be transformed into its additive counterpart $\varepsilon$ in order to perform the gradient step in the original additive diffusion process domain. We obtain the additive noise by subtracting the modulated predicted codeword $\text{sign}(\hat{x})$ from the noisy signal, such that $\hat{\varepsilon} = y - \text{sign}(\hat{x}) = y - \text{sign}(\hat{\tilde{\varepsilon}}y)$. Therefore, following

| **Algorithm 1:** DDECC training procedure. | **Algorithm 2:** DDECC sampling procedure |
|---|---|
| 1: $x_0 \in C$ 
 2: Input: Parity check matrix $H$, noise schedule $\beta_1, ..., \beta_T$ 
 3: **repeat** 
 4:     $t \sim \mathcal{U}(\{1, ..., T\})$ 
 5:     $\varepsilon \sim \mathcal{N}(0, I)$ 
 6:     $x_t = x_0 + \sqrt{\bar{\beta}_t}\varepsilon = x_0\tilde{\varepsilon}$ 
 7:     Take gradient descent step on: 
       $\mathrm{BCE}(\epsilon_\theta(x_t, e_t), \mathrm{bin}(\tilde{\varepsilon}))$ 
 8: **until** converged | 1: Input: Parity check matrix $H$, channel's output $y$ 
 2: **for** $n - k$ iterations **do** 
 3:     $\gamma = e(\mathrm{bin}(y))$ 
 4:     **if** $\gamma = 0$ **then** 
 5:       **return** $\mathrm{bin}(y)$ 
 6:     $\hat{\tilde{\varepsilon}} = \varepsilon_\theta(y, \gamma) \,; \hat{\varepsilon} = y - \mathrm{sign}(\hat{\tilde{\varepsilon}}y)$ 
 7:     Get $\lambda$ according to Eq. 14 
 8:     $y = y - \lambda\{\bar{\beta}_\gamma\beta\gamma^{-1}/(\bar{\beta}_\gamma + \beta_\gamma)\hat{\varepsilon}$ 
 9: **return** $\mathrm{bin}(y)$ |

Eq. 9, at inference time the reverse process is given by

$$x_{t-1} = x_t - \frac{\sqrt{\bar{\beta}_t}\beta_t}{\bar{\beta}_t + \beta_t}\big(x_t - \mathrm{sign}(x_t\hat{\tilde{\varepsilon}})\big) \quad = x_t - \frac{\sqrt{\bar{\beta}_t}\beta_t}{\bar{\beta}_t + \beta_t}\big(x_t - \mathrm{sign}(x_t\epsilon_\theta(x_t, e_t))\big) \tag{13}$$

The inference procedure is defined in Alg.2. If the syndrome is non-zero, we predict the multiplicative noise, extract the corresponding additive noise, and perform the reverse step. We illustrate in Fig.2 the reverse diffusion dynamics (gradient field) for a $(3, 1)$ repetition code, i.e., $G = (1, 1, 1)$.

### 4.4 Syndrome-based Line Search for Reverse Diffusion Step Size

One major limitation of the generative neural diffusion process is the large number of diffusion steps required - generally a thousand - in order to generate high-quality samples. Several methods proposed faster sampling procedures in order to accelerate data generation via schedule subsampling or step size correction (Nichol & Dhariwal, 2021b; San-Roman et al., 2021). In our configuration, one can assess the quality of the denoised signal via the value of its syndrome, i.e., the number of parity check errors, while a zero syndrome means a valid codeword.

Therefore, we propose to find the optimal step size $\lambda$ by solving the following optimization problem

$$\lambda^* = \underset{\lambda \in \mathbb{R}^+}{\arg\min} \|s\big(x_t - \lambda\frac{\sqrt{\bar{\beta}_t}\beta t}{\bar{\beta}_t + \beta_t}\hat{\varepsilon}\big)\|_1, \tag{14}$$

where $s(\cdot)$ denotes the syndrome computed over $GF(2)$ as in Eq. 2.

While many line-search (LS) methods exist in numerical optimization (Nocedal & Wright, 2006), since the objective is highly non-differentiable, we suggest adopting a grid search procedure such that the search space becomes restricted to $\lambda \in I$ where $I$ is a predefined discrete segment. This parallelizable procedure reduces the number of iterations by a sizable factor, as shown in Section 5. Details regarding the grid-search procedure are discussed in Appendix C.

**Architecture and Training** The state-of-the-art ECCT architecture of Choukroun & Wolf (2022) is used as $\epsilon_\theta$. In this architecture, the capacity of the model is defined according to the chosen embedding dimension $d$ and the number of self-attention layers $N$. In order to condition the network by the number of parity errors $e_t \in \{0, \ldots, n - k\}$, we employ a $d$ dimensional one hot encoding multiplied via Hadamard product with the initial elements' embedding of the ECCT. Denoting the ECCT's embedding of the $i$ element as $\phi_i$, the new embedding is defined as $\tilde{\phi}_i = \phi_i \odot \psi(e_t), \forall i$, where $\psi$ denotes the $n - k$ one hot embedding. As a transformation of the syndrome, $e_t$ remains also invariant to the codeword. Additional details on the DDECCT architecture are given in Appendix F.

The discrete grid search of $\lambda$ is uniformly sampled over $I = [1, 20]$ with 20 samples, in order to find the optimal step size. A denser or a code adaptive sampling may improve the results, according to a predefined computation-speed trade-off. We show the distribution of optimal $\lambda$ in Appendix C.

The Adam optimizer (Kingma & Ba, 2014) is used with 128 samples per mini-batch, for 2000 epochs, with 1000 mini-batches per epoch. The noise scheduling is constant and set to $\beta_t = 0.01, \forall t$. An extended discussion regarding the $\beta$ scheduler can be found in Appendix G. We initialized the learning rate to $10^{-4}$ coupled with a cosine decay scheduler down to $5 \cdot 10^{-6}$ at the end of training. No warmup (Xiong et al., 2020) was employed.

Table 1: A comparison of the negative natural logarithm of Bit Error Rate (BER) for three normalized SNR values (4,5,6) of our method with literature baselines. Higher is better. The best results are in **bold**, second best underlined.
BP-based results are obtained after $L = 5$ BP iterations in the first row (i.e. 10-layer neural network) and *at convergence* results in the second row are obtained after $L = 50$ BP iterations (i.e., 100-layer neural network). Our performance is presented for six different architectures: for $N = \{2, 6\}$ and $d = \{32, 64, 128\}$. The presented results are obtained with the LS procedure.

| Method | BP | | | ARBP | | | ECCT N=2 | | | ECCT N=6 | | | Ours N=2 | | | Ours N=6 | | |
|---|---|---|---|---|---|---|---|---|---|---|---|---|---|---|---|---|---|---|
| | 4 | 5 | 6 | 4 | 5 | 6 | 4 | 5 | 6 | 4 | 5 | 6 | 4 | 5 | 6 | 4 | 5 | 6 |
| Polar(64,32) | 3.52 | 4.04 | 4.48 | 4.77 | 6.30 | 8.19 | 4.27 | 5.44 | 6.95 | 5.71 | 7.63 | 9.94 | 5.99 | 8.16 | 10.90 | 6.76 | 9.14 | 12.31 |
| | 4.26 | 5.38 | 6.50 | 5.57 | 7.43 | 9.82 | 4.57 | 5.86 | 7.50 | 6.48 | 8.60 | 11.43 | 6.23 | 8.52 | 11.23 | 6.90 | 9.43 | **12.85** |
| | | | | | | | 4.87 | 6.2 | 7.93 | 6.99 | 9.44 | 12.32 | 6.59 | 8.95 | 11.91 | **6.93** | **9.51** | 12.79 |
| Polar(64,48) | 4.15 | 4.68 | 5.31 | 5.25 | 6.96 | 9.00 | 4.92 | 6.46 | 8.41 | 5.82 | 7.81 | 10.24 | 5.55 | 7.67 | 10.08 | 5.98 | 8.02 | 10.94 |
| | 4.74 | 5.94 | 7.42 | 5.41 | 7.19 | 9.30 | 5.14 | 6.78 | 8.9 | 6.15 | 8.20 | 10.86 | 5.74 | 7.85 | 10.40 | 5.98 | **8.26** | **11.13** |
| | | | | | | | 5.36 | 7.12 | 9.39 | 6.36 | 8.46 | 11.09 | 5.77 | 7.94 | 10.64 | **6.00** | 8.24 | 10.98 |
| Polar(128,64) | 3.38 | 3.80 | 4.15 | 4.02 | 5.48 | 7.55 | 3.51 | 4.52 | 5.93 | 4.47 | 6.34 | 8.89 | 5.37 | 7.75 | 10.51 | 6.34 | 9.26 | 12.77 |
| | 4.10 | 5.11 | 6.15 | 4.84 | 6.78 | 9.30 | 3.83 | 5.16 | 7.04 | 5.12 | 7.36 | 10.48 | 5.97 | 8.52 | 11.72 | 7.24 | 10.70 | 14.56 |
| | | | | | | | 4.04 | 5.52 | 7.62 | 5.92 | 8.64 | 12.18 | 6.50 | 9.23 | 12.37 | **9.11** | **12.90** | **16.30** |
| Polar(128,86) | 3.80 | 4.19 | 4.62 | 4.81 | 6.57 | 9.04 | 4.30 | 5.58 | 7.34 | 5.36 | 7.45 | 10.22 | 5.61 | 7.76 | 10.42 | 6.52 | 9.21 | 12.64 |
| | 4.49 | 5.65 | 6.97 | 5.39 | 7.37 | 10.13 | 4.49 | 5.90 | 7.75 | 5.75 | 8.16 | 11.29 | 5.99 | 8.19 | 11.00 | 7.09 | 10.20 | 13.84 |
| | | | | | | | 4.75 | 6.25 | 8.29 | 6.31 | 9.01 | 12.45 | 6.27 | 8.64 | 11.61 | **7.60** | **10.81** | **15.17** |
| Polar(128,96) | 3.99 | 4.41 | 4.78 | 4.92 | 6.73 | 9.30 | 4.56 | 5.98 | 7.93 | 5.39 | 7.62 | 10.45 | 5.60 | 7.83 | 10.56 | 6.46 | 9.41 | 12.52 |
| | 4.61 | 5.79 | 7.08 | 5.27 | 7.44 | 10.2 | 4.69 | 6.20 | 8.30 | 5.88 | 8.33 | 11.49 | 5.95 | 8.42 | 11.38 | 6.83 | 9.99 | **13.36** |
| | | | | | | | 4.88 | 6.58 | 8.93 | 6.31 | 9.12 | 12.47 | 6.26 | 8.94 | 12.01 | **7.16** | **10.3** | 13.19 |
| LDPC(49,24) | 5.30 | 7.28 | 9.88 | 6.05 | 8.13 | 11.68 | 4.51 | 6.07 | 8.11 | 5.74 | 8.13 | 11.30 | 5.27 | 7.38 | 10.23 | 5.87 | 8.22 | 11.56 |
| | 6.23 | 8.19 | 11.72 | 6.58 | 9.39 | 12.39 | 4.58 | 6.18 | 8.46 | 5.91 | 8.42 | 11.90 | 5.31 | 7.35 | 10.40 | 5.84 | 8.29 | 11.85 |
| | | | | | | | 4.71 | 6.38 | 8.73 | **6.13** | 8.71 | **12.10** | 5.36 | 7.39 | 10.41 | 6.01 | **8.74** | 11.92 |
| LDPC(121,60) | 4.82 | 7.21 | 10.87 | 5.22 | 8.31 | 13.07 | 3.88 | 5.51 | 8.06 | 4.98 | 7.91 | 12.70 | 4.48 | 6.95 | 10.65 | 5.25 | 8.43 | 13.80 |
| | - | - | - | - | - | - | 3.89 | 5.55 | 8.16 | 5.02 | 7.94 | 12.72 | 4.56 | 7.02 | 10.64 | 5.32 | 8.69 | 13.82 |
| | | | | | | | 3.93 | 5.66 | 8.51 | 5.17 | 8.31 | 13.30 | 4.46 | 6.92 | 10.76 | **5.38** | **8.73** | **14.17** |
| LDPC(121,70) | 5.88 | 8.76 | 13.04 | 6.45 | 10.01 | 14.77 | 4.63 | 6.68 | 9.73 | 6.11 | 9.62 | 15.10 | 5.41 | 8.22 | 12.22 | 6.49 | 10.39 | 15.43 |
| | - | - | - | - | - | - | 4.64 | 6.71 | 9.77 | 6.28 | 10.12 | 15.57 | 5.52 | 8.47 | 12.63 | 6.64 | 10.65 | 16.21 |
| | | | | | | | 4.67 | 6.79 | 9.98 | 6.40 | 10.21 | 16.11 | 5.55 | 8.51 | 12.81 | **6.79** | **11.13** | **16.93** |
| LDPC(121,80) | 6.66 | 9.82 | 13.98 | 7.22 | 11.03 | 15.90 | 5.27 | 7.59 | 10.08 | 6.92 | 10.74 | 15.10 | 6.12 | 9.38 | 13.25 | 7.68 | **12.19** | **17.83** |
| | - | - | - | - | - | - | 5.29 | 7.63 | 10.90 | 7.17 | 11.21 | 16.31 | 6.26 | 9.41 | 13.41 | 7.39 | 11.46 | 17.65 |
| | | | | | | | 5.30 | 7.65 | 11.03 | 7.41 | 11.51 | 16.44 | 6.26 | 9.41 | 13.46 | **7.71** | 12.17 | 17.55 |
| MacKay(96,48) | 6.84 | 9.40 | 12.57 | 7.43 | 10.65 | 14.65 | 4.95 | 6.67 | 8.94 | 6.88 | 9.86 | 13.40 | 6.18 | 8.63 | 11.53 | 7.86 | 11.61 | 15.51 |
| | - | - | - | - | - | - | 5.04 | 6.80 | 9.23 | 7.10 | 10.12 | 14.21 | 6.28 | 8.8 | 11.78 | 7.93 | 11.65 | 15.51 |
| | | | | | | | 5.17 | 7.07 | 9.64 | 7.38 | 10.72 | 14.83 | 6.31 | 8.83 | 12.03 | **8.12** | **11.88** | **15.93** |
| CCSDS(128,64) | 6.55 | 9.65 | 13.78 | 7.25 | 10.99 | 16.36 | 4.35 | 6.01 | 8.30 | 6.34 | 9.80 | 14.40 | 5.79 | 8.48 | 12.24 | 7.28 | 11.66 | 17.02 |
| | - | - | - | - | - | - | 4.41 | 6.09 | 8.49 | 6.65 | 10.40 | 15.46 | 5.81 | 8.79 | 12.29 | 7.55 | 12.01 | 17.62 |
| | | | | | | | 4.59 | 6.42 | 9.02 | 6.88 | 10.90 | 15.90 | 6.22 | 9.72 | 13.71 | **7.81** | **12.48** | **17.66** |
| BCH(63,36) | 3.72 | 4.65 | 5.66 | 4.33 | 5.94 | 8.21 | 3.79 | 4.87 | 6.35 | 4.42 | 5.91 | 8.01 | 4.71 | 6.45 | 8.72 | 5.01 | 6.84 | 9.30 |
| | 4.03 | 5.42 | 7.26 | 4.57 | 6.39 | 8.92 | 4.05 | 5.28 | 7.01 | 4.62 | 6.24 | 8.44 | 4.84 | 6.65 | 9.01 | 5.07 | 7.02 | 9.85 |
| | | | | | | | 4.21 | 5.50 | 7.25 | 4.86 | 6.65 | 9.10 | **5.19** | **7.27** | 9.82 | **5.19** | 7.10 | **9.96** |
| BCH(63,45) | 4.08 | 4.96 | 6.07 | 4.80 | 6.43 | 8.69 | 4.47 | 5.88 | 7.81 | 5.16 | 7.02 | 9.75 | 5.12 | 7.16 | 9.95 | 5.49 | 7.71 | 10.86 |
| | 4.36 | 5.55 | 7.26 | 4.97 | 6.90 | 9.41 | 4.66 | 6.16 | 8.17 | 5.41 | 7.49 | 10.25 | 5.33 | 7.49 | 10.18 | 5.60 | **8.02** | 11.05 |
| | | | | | | | 4.79 | 6.39 | 8.49 | 5.60 | 7.79 | 10.93 | 5.41 | 7.61 | 10.46 | **5.61** | 7.94 | **11.36** |
| BCH(63,51) | 4.34 | 5.29 | 6.35 | 4.95 | 6.69 | 9.18 | 4.60 | 6.05 | 8.05 | 5.20 | 7.08 | 9.65 | 5.09 | 7.08 | 9.87 | 5.35 | 7.49 | 10.38 |
| | 4.5 | 5.82 | 7.42 | 5.17 | 7.16 | 9.53 | 4.78 | 6.34 | 8.49 | 5.46 | 7.57 | 10.51 | 5.19 | 7.23 | 10.20 | 5.39 | 7.48 | 10.53 |
| | | | | | | | 5.01 | 6.72 | 9.03 | **5.47** | 7.59 | 10.62 | 5.33 | 7.29 | 10.13 | **5.47** | **7.66** | **10.73** |

Training and experiments were performed on a 12GB Titan V GPU. The total training time ranged from 12 to 24 hours depending on the code length, and no optimization of the self-attention mechanism was employed. Per epoch, the training time was in the range of 19-40 and 40-102 seconds for the $N = 2, 6$ architectures, respectively.

## 5 EXPERIMENTS

To evaluate our method, we train the proposed architecture with three classes of linear block codes: Low-Density Parity Check (LDPC) codes (Gallager, 1962), Polar codes (Arikan, 2008) and Bose–Chaudhuri–Hocquenghem (BCH) codes (Bose & Ray-Chaudhuri, 1960). All parity check matrices are taken from Helmling et al. (2019).

The proposed architecture is defined solely by the number of encoder layers $N$ and the dimension of the embedding $d$. We compare our method with the BP algorithm (Pearl, 1988), the recent Autore-

Table 2: A comparison between the line search procedure and the regular reverse diffusion. The $\Delta$ column denotes the difference between the logarithm of Bit Error (BER) for three normalized SNR values (i.e., $\Delta = -\log(\mathrm{BER}_{LS}) + \log(\mathrm{BER}_{Reg})$).

The other columns represent the mean and standard deviation of the number of iterations of the reverse process until convergence, i.e., convergence to zero syndrome.

| Method | $\Delta$ N=2 | | | $\Delta$ N=6 | | | #It. Reg. N=2 | | | #It Reg. N=6 | | | #It. LS N=2 | | | #It LS N=6 | | |
|---|---|---|---|---|---|---|---|---|---|---|---|---|---|---|---|---|---|---|
| | 4 | 5 | 6 | 4 | 5 | 6 | 4 | 5 | 6 | 4 | 5 | 6 | 4 | 5 | 6 | 4 | 5 | 6 |
| Polar(64,48) | -0.02 | 0.11 | 0.16 | 0.01 | 0.06 | 0.47 | 5.9±4.9 | 3.3±3.8 | 1.5±2.7 | 5.7±4.6 | 3.2±3.8 | 1.5±2.7 | 1.4±2.3 | 0.7±1.0 | 0.4±0.5 | 1.2±1.7 | 0.7±0.9 | 0.4±0.6 |
| | 0.01 | 0.14 | 0.30 | -0.03 | 0.12 | 0.43 | 5.8±4.8 | 3.2±3.8 | 1.5±2.7 | 5.7±4.6 | 3.2±3.7 | 1.5±2.7 | 1.3±2.0 | 0.7±0.9 | 0.4±0.5 | 0.9±1.0 | 0.6±0.6 | 0.4±0.5 |
| | 0.00 | 0.16 | 0.40 | -0.01 | 0.07 | 0.71 | 5.8±4.7 | 3.2±3.8 | 1.5±2.7 | 5.7±4.6 | 3.2±3.7 | 1.5±2.7 | 1.2±1.8 | 0.7±0.8 | 0.4±0.5 | 1.0±1.0 | 0.6±0.6 | 0.4±0.5 |
| Polar(128,86) | -0.10 | -0.11 | -0.20 | -0.20 | -0.13 | -0.21 | 16.5±10.9 | 9.1±7.3 | 4.7±4.9 | 13.4±7.5 | 8.3±5.8 | 4.6±4.6 | 4.0±9.1 | 1.4±3.4 | 0.8±1.0 | 2.9±4.4 | 1.6±2.1 | 1.0±1.2 |
| | -0.06 | -0.04 | 0.00 | -0.30 | -0.34 | -0.33 | 15.5±10.0 | 8.8±6.8 | 4.6±4.8 | 13.1±6.9 | 8.3±5.7 | 4.6±4.6 | 3.4±8.3 | 1.3±3.2 | 0.8±0.9 | 1.3±2.4 | 1.0±0.6 | 0.7±0.5 |
| | -0.09 | -0.06 | -0.10 | -0.32 | -0.21 | -0.40 | 14.7±9.2 | 8.6±6.4 | 4.6±4.7 | 13.0±6.6 | 8.3±5.7 | 4.6±4.6 | 2.7±6.8 | 1.2±2.3 | 0.8±0.7 | 1.2±1.9 | 0.9±0.5 | 0.7±0.5 |
| Polar(128,96) | -0.10 | -0.13 | -0.20 | -0.11 | 0.04 | -0.16 | 12.6±8.8 | 6.5±5.9 | 3.1±3.9 | 10.5±6.4 | 6.1±5.0 | 3.1±3.8 | 3.6±7.4 | 1.2±2.7 | 0.6±0.8 | 2.2±3.4 | 1.2±1.4 | 0.7±0.8 |
| | -0.09 | -0.10 | -0.10 | -0.19 | 0.13 | 0.14 | 11.8±8.0 | 6.3±5.5 | 3.1±3.8 | 10.32±6.11 | 6.1±4.9 | 3.1±3.8 | 2.7±6.0 | 1.1±2.0 | 0.6±0.6 | 1.2±2.0 | 0.9±0.5 | 0.6±0.5 |
| | -0.16 | -0.12 | -0.20 | -0.15 | 0.00 | -0.13 | 11.1±7.3 | 6.2±5.2 | 3.1±3.8 | 10.2±5.9 | 6.1±4.9 | 3.1±3.8 | 2.0±4.4 | 1.0±1.3 | 0.6±0.5 | 1.1±1.5 | 0.9±0.5 | 0.6±0.5 |
| LDPC(49,24) | 0.06 | 0.03 | 0.23 | -0.05 | -0.31 | -0.16 | 11.5±7.0 | 7.4±5.7 | 4.4±4.5 | 10.9±6.3 | 7.3±5.4 | 4.4±4.5 | 2.2±5.1 | 1.0±1.9 | 0.7±0.7 | 2.2±3.6 | 1.3±1.5 | 0.8±0.8 |
| | 0.05 | -0.06 | 0.20 | -0.13 | -0.12 | -0.20 | 11.4±7.0 | 7.4±5.7 | 4.4±4.5 | 10.9±6.3 | 7.3±5.4 | 4.4±4.5 | 2.1±4.9 | 1.0±1.9 | 0.7±0.6 | 1.5±3.5 | 0.9±1.1 | 0.7±0.5 |
| | 0.07 | -0.10 | 0.20 | -0.12 | -0.13 | -0.41 | 11.4±7.0 | 7.4±5.6 | 4.4±4.5 | 10.9±6.3 | 7.3±5.4 | 4.4±4.5 | 2.1±4.8 | 1.0±1.9 | 0.7±0.6 | 1.4±3.4 | 0.9±1.1 | 0.7±0.5 |
| LDPC(121,80) | -0.23 | -0.03 | -0.40 | -0.33 | -0.46 | -0.95 | 12.5±7.9 | 7.3±5.0 | 4.0±3.8 | 11.4±5.6 | 7.2±4.7 | 4.0±3.8 | 2.7±7.5 | 1.0±1.6 | 0.7±0.5 | 1.2±3.0 | 0.9±0.4 | 0.7±0.4 |
| | -0.13 | -0.10 | -0.30 | -0.15 | -0.42 | 0.81 | 12.5±7.9 | 7.3±4.9 | 4.0±3.8 | 11.4±5.8 | 7.2±4.7 | 4.0±3.8 | 2.7±7.7 | 1.0±1.7 | 0.7±0.5 | 1.4±3.7 | 0.9±0.6 | 0.7±0.4 |
| | -0.10 | -0.17 | -0.20 | -0.28 | -0.21 | -0.27 | 12.4±7.82 | 7.2±4.9 | 4.0±3.8 | 11.4±5.6 | 7.2±4.7 | 4.0±3.8 | 3.1±6.9 | 1.3±1.7 | 0.8±0.6 | 1.3±3.2 | 0.9±0.5 | 0.7±0.4 |
| MacKay(96,48) | -0.23 | -0.16 | -0.50 | -0.09 | 0.29 | 0.00 | 15.3±7.9 | 10.2±5.5 | 6.4±4.5 | 14.4±5.7 | 10.0±5.0 | 6.4±4.4 | 2.8±8.1 | 1.2±2.6 | 0.9±0.7 | 2.2±2.9 | 1.5±0.9 | 1.1±0.6 |
| | -0.20 | -0.14 | -0.30 | -0.17 | -0.21 | 0.00 | 15.3±7.8 | 10.2±5.4 | 6.4±4.5 | 14.3±5.7 | 10.0±5.0 | 6.4±4.4 | 2.6±7.8 | 1.2±2.5 | 0.9±0.7 | 1.3±2.6 | 1.0±0.5 | 0.9±0.3 |
| | -0.19 | -0.17 | -0.20 | -0.19 | -0.33 | -0.13 | 15.2±7.7 | 10.2±5.4 | 6.4±4.5 | 14.3±5.6 | 10.0±5.0 | 6.4±4.4 | 2.6±7.6 | 1.2±2.4 | 0.9±0.6 | 1.2±2.3 | 1.0±0.4 | 0.9±0.3 |
| CCSDS(128,64) | -0.19 | -0.58 | -0.40 | -0.31 | -0.36 | 0.52 | 20.8±11.4 | 13.1±6.3 | 8.4±4.9 | 18.2±6.6 | 12.8±5.4 | 8.4±4.9 | 4.7±13.4 | 1.4±4.1 | 1.0±0.8 | 1.7±4.7 | 1.1±0.7 | 1.0±0.3 |
| | -0.21 | -0.30 | -0.60 | -0.23 | -0.42 | 0.31 | 20.6±11.2 | 13.1±6.2 | 8.4±4.9 | 18.1±6.4 | 12.8±5.4 | 8.4±4.9 | 4.4±12.8 | 1.3±3.3 | 1.0±0.7 | 1.6±4.1 | 1.1±0.6 | 1.0±0.3 |
| | -0.27 | -0.42 | -0.40 | -0.26 | -0.11 | 0.37 | 20.6±11.0 | 13.1±6.2 | 8.4±4.9 | 18.1±6.3 | 12.8±5.4 | 8.4±4.9 | 4.2±12.3 | 1.3±3.4 | 1.0±0.6 | 1.6±3.7 | 1.1±0.5 | 1.0±0.3 |
| BCH(63,36) | -0.01 | 0.02 | -0.04 | 0.01 | 0.00 | 0.09 | 12.6±8.0 | 7.8±6.7 | 4.3±5.1 | 11.9±7.5 | 7.6±6.4 | 4.3±5.0 | 3.7±7.2 | 1.4±3.4 | 0.7±1.3 | 4.3±6.8 | 2.0±3.6 | 1.0±1.8 |
| | -0.01 | 0.02 | 0.13 | -0.11 | -0.05 | 0.03 | 12.4±7.9 | 7.7±6.6 | 4.3±5.1 | 11.8±7.4 | 7.5±6.3 | 4.3±5.0 | 3.3±6.8 | 1.3±3.1 | 0.7±1.1 | 2.5±5.6 | 1.1±2.4 | 0.7±0.8 |
| | 0.04 | 0.16 | 0.23 | -0.11 | -0.02 | 0.04 | 12.1±7.6 | 7.6±6.5 | 4.3±5.0 | 11.7±7.3 | 7.5±6.3 | 4.3±5.0 | 2.5±5.5 | 1.1±2.2 | 0.7±0.8 | 2.5±5.6 | 1.1±2.3 | 0.7±0.8 |
| BCH(63,51) | 0.04 | 0.28 | 0.94 | 0.09 | 0.42 | 1.16 | 4.8±4.1 | 2.6±3.4 | 1.2±2.4 | 4.7±4.0 | 2.6±3.3 | 1.2±2.4 | 1.8±2.9 | 0.7±1.3 | 0.3±0.6 | 1.6±2.6 | 0.7±1.2 | 0.3±0.6 |
| | 0.06 | 0.31 | 1.13 | 0.06 | 0.28 | 1.19 | 4.8±4.1 | 2.6±3.4 | 1.2±2.4 | 4.7±4.0 | 2.6±3.4 | 1.2±2.4 | 1.6±2.7 | 0.7±1.2 | 0.3±0.5 | 1.3±2.1 | 0.6±1.0 | 0.3±0.5 |
| | 0.04 | 0.34 | 1.02 | 0.02 | 0.34 | 1.04 | 4.8±4.1 | 2.6±3.4 | 1.2±2.4 | 4.7±4.0 | 2.6±3.4 | 1.2±2.3 | 1.6±2.6 | 0.7±1.1 | 0.3±0.5 | 1.4±2.3 | 0.6±1.0 | 0.3±0.5 |

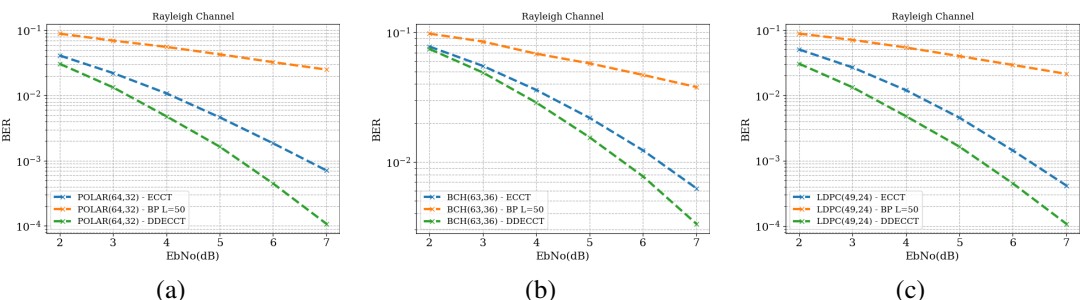

(a)  (b)  (c)

Figure 4: BER comparison between the ECCT $N = 6, d = 32$ and the proposed DDECCT, for the Rayleigh fading channel for (a) Polar(64,32), (b) BCH(63,36), and (c) LDPC(49,24) codes.

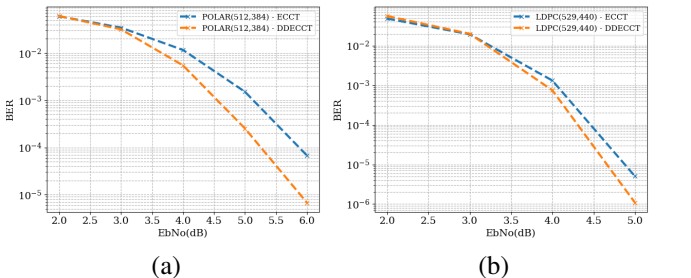
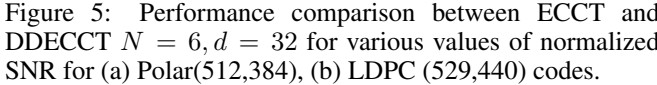

(a)  (b)

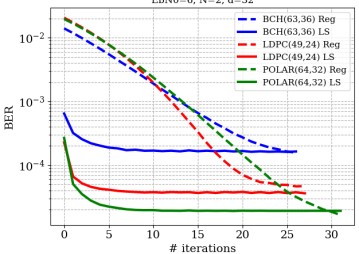

Figure 5: Performance comparison between ECCT and DDECCT $N = 6, d = 32$ for various values of normalized SNR for (a) Polar(512,384), (b) LDPC (529,440) codes.

Figure 6: BER vs the number of iterations (up to $n - k$) for regular and line search reverse diffusion.

gressive hyper-network BP of Nachmani & Wolf (2021) (AR BP) and the SOTA ECCT (Choukroun & Wolf, 2022). Since our decoder is based on the ECCT, the contribution of the diffusion model scheme is pertinent in comparing our results with ECCT since they have similar architectures and capacities. Our method's overhead over ECCT is by a factor of the number of diffusion steps $L$, i.e., a complexity of $\mathcal{O}(LN(d^2(2n - k) + hd))$, where $h$ is the complexity of the self-attention module.

We refer the reader to Choukroun & Wolf (2022) for a detailed complexity analysis of the ECCT. Details about the computational overhead of the DDECCT are given in Appendix F. Note that LDPC codes are designed specifically for BP-based decoding (Richardson et al., 2001).

The results are reported as bit error rates (BER) for different normalized SNR values ($Eb/N_0$). We follow the testing benchmark of (Nachmani & Wolf, 2019; Choukroun & Wolf, 2022). During testing, our decoder decodes at least $10^5$ random codewords, to obtain at least $500$ frames with errors at each SNR value. All baseline results were obtained from the corresponding papers.

The results are reported in Tab. 1, where we present the negative natural logarithm of the BER. For each code, we present the results of the BP-based competing methods for 5 and 50 iterations (first and second rows), corresponding to a neural network with 10 and 100 layers, respectively. As in (Choukroun & Wolf, 2022), our framework's performance with Line Search (LS) as described in Section 4.4 is evaluated for six different architectures, with $N = \{2, 6\}$ and $d = \{32, 64, 128\}$, respectively (first to third rows). BER plots with respect to the SNR are given in Appendix I.

As can be seen, our approach outperforms the current SOTA results of ECCT by extremely large margins on several codes, at a fraction of the capacity. Especially for shallow models, the difference can be an order of magnitude. Performance is closer with short high-rate codes, for which ECCT performance is already very high. We present in Figure 5 the performance of the proposed DDECCT on larger codes. As can be seen, DDECCT can learn to efficiently decode larger codes and outperforms ECCT. A separate comparison to the non-neural SCL Polar decoder of Tal & Vardy (2015) is given in Appendix H, demonstrating the need to train bigger architectures in order to surpass this specialized decoder.

We present in Table 2 the difference in accuracy $\Delta$ between the line search procedure and the regular reverse diffusion. We also present convergence statistics (mean and standard deviation of the number of iterations) for the regular reverse diffusion and the line search procedure. The full table with the statistics for all of the codes is given in Appendix E. Evidently, the line search procedure enables extremely fast convergence, requiring as little as one iteration for high SNR. Note that we measure the number of iterations required to reach a syndrome of zero failed checks. We do not apply early stopping to the decoding, which could reduce the average number of iterations even further if the decoder stagnates and does not converge to zero syndrome.

**Non-Gaussian Channel** We test our framework on a non-Gaussian Rayleigh fading channel, which is often used for simulating the propagation environment of a signal, e.g., for wireless devices. In this fading model, the transmission of the codeword $x \in \{0, 1\}^n$ is defined as $y = hx_s + z$, where $h$ is an $n$-dimensional i.i.d. Rayleigh-distributed vector with a scale parameter $\alpha$, and $z \sim \mathcal{N}(0, \sigma^2 I_n)$.

In our simulations, we assume a *high* scale $\alpha = 1$ in order to easily compare and reproduce the results, while the level of Gaussian noise and the testing procedure remain the same as described in the paper. The overall variance of the transmitted codeword $y$ in the Rayleigh channel is roughly *twice* the AWGN's on the tested SNR range. The results are presented in Figure 4. As can be observed, our method is still able to learn to decode, even under these very noisy fading channels.

**BER evolution through iteration/time** We illustrate in Figure 6 the denoising process for several codes. We show how the BER decreases with time for the regular proposed method and the augmented line search procedure. We can observe the very fast convergence of the line search approach. We further provide in Appendix D the performance of the proposed framework for one, two and three iteration steps. We can see that LS enables outperforming the original ECCT, even with *one step* only.

## 6 CONCLUSIONS

We present a novel denoising diffusion method for the decoding of algebraic block codes. It is based on an adapted diffusion process that simulates the channel corruption we wish to reverse. The method makes use of the syndrome as a conditioning signal and employs a line-search procedure to control the step size. Since it inherits the iterative nature of the underlying process, both training and deployment are extremely efficient. Even with very low-capacity networks, the proposed approach outperforms existing neural decoders by sizable margins for a broad range of code families.

ACKNOWLEDGMENTS

This project has received funding from the European Research Council (ERC) under the European Unions Horizon 2020 research, innovation program (grant ERC CoG 725974). This work was further supported by a grant from the Tel Aviv University Center for AI and Data Science (TAD). The contribution of the first author is part of a Ph.D. thesis research conducted at Tel Aviv University.

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

## A  UNSCALED DIFFUSION DERIVATION

According to Bayes' rule,

$$
\begin{aligned}
q(x_{t-1}|x_t, x_0) &= q(x_t|x_{t-1}, x_0)\frac{q(x_{t-1}|x_0)}{q(x_t|x_0)} \\
&\propto \exp\Big(-\frac{1}{2}\big(\frac{(x_t - x_{t-1})^2}{\beta_t} + \frac{(x_{t-1} - x_0)^2}{1 - \bar{\beta}_t} - \frac{(x_t - x_0)^2}{1 - \bar{\beta}_t}\big)\Big) \\
&= \exp\Big(-\frac{1}{2}\big((\frac{1}{\beta_t} + \frac{1}{\bar{\beta}_t})x_{t-1}^2 - (\frac{2}{\beta_t}x_t - \frac{2}{\bar{\beta}_t}x_0)x_{t-1} + C(x_t, x_0)\big)\Big)
\end{aligned}
\tag{15}
$$

where $C(x_t, x_0)$ represents the constant term of the second-order equation. Following the standard Gaussian density function, the mean and variance can be parameterized as follows

$$
\begin{aligned}
\tilde{\beta}_t &= \Big(\frac{1}{\beta_t} + \frac{1}{\bar{\beta}_t}\Big)^{-1} \\
\tilde{\mu}_t(x_t, x_0) &= (\frac{1}{\beta_t}x_t + \frac{1}{\bar{\beta}_t}x_0)/\tilde{\beta}_t \\
&= \frac{\bar{\beta}_t}{\bar{\beta}_t + \beta_t}x_t + \frac{\beta_t}{\bar{\beta}_t + \beta_t}x_0 \\
&= \frac{\bar{\beta}_t}{\bar{\beta}_t + \beta_t}x_t + \frac{\beta_t}{\bar{\beta}_t + \beta_t}(x_t - \sqrt{\bar{\beta}_t}\varepsilon) \\
&= x_t - \frac{\sqrt{\bar{\beta}_t}\beta_t}{\bar{\beta}_t + \beta_t}\varepsilon.
\end{aligned}
\tag{16}
$$

## B  FORWARD DIFFUSION PROCESS VISUALIZATION

We provide a visualization of the forward diffusion process as described in Section 4.1, using the three dimensional repetition code as discussed in Section 4.3.

Figure 7 presents random diffusion processes from one of the two valid codewords through time. As can be seen, there is a migration from the valid codewords (depicted by either a blue or a red cross) to the vicinity of invalid words (black crosses).

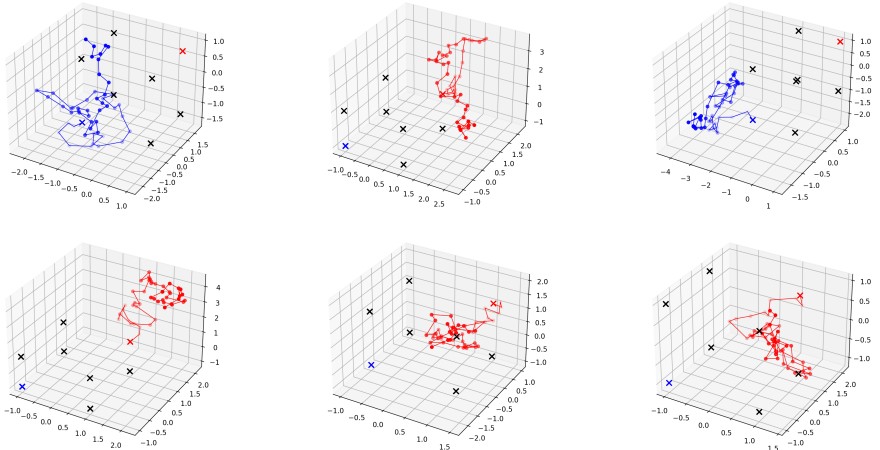

Figure 7: Visualization of the forward diffusion process. The two valid codewords are represented as a blue cross and a red cross. The other six words are denoted by black crosses.

## C    LINE SEARCH HISTOGRAMS AND COMPLEXITY

### C.1    HISTOGRAMS

Figure 8 presents the distribution of the optimal step size $\lambda$ for several codes. Each code presents a different distribution of the optimal step sizes, as can be seen from the high variance of the x-axis.

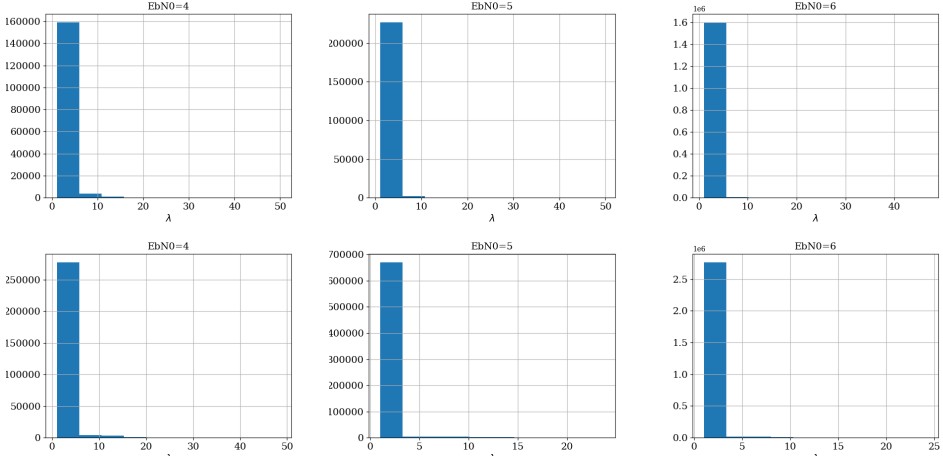

Figure 8: Histograms of optimal $\lambda$ values on the test set for the POLAR(128,96) code (first row) and the LDP(49,24) code (second row), for $Eb/N_0 = \{4, 5, 6\}$ corresponding to the left to right histograms, respectively. The grid search was sampled uniformly, taking 300 samples from the presented range.

### C.2    COMPLEXITY OVERHEAD OF THE LINE SEARCH PROCEDURE

The computation of the syndrome consists of a series of efficient binary operations (xor) inducing a computational complexity that is proportional to the density of the code.

The line search consists of the parallel computation of the syndrome, over the multiple words obtained for different $\lambda$ values sampled over a predefined grid. Thus, the time complexity of the line-search procedure can be reduced to the very efficient computation of the syndrome which can be assumed as constant. Without parallelization, the complexity is linear with the grid size, but the overall process remains extremely efficient.

# D  DDECCT PERFORMANCE WITH FEW ITERATIONS

Table 3 presents the performance of the proposed framework for one, two, and three iteration steps with the regular reverse diffusion method and with the proposed line-search approach. As can be seen, the line-search approach improves over the regular reverse diffusion by orders of magnitude.

Table 3: Negative natural logarithm of BER by number of iterations for N=2 models for the regular and the LS diffusion methods. Higher is better. The (column) average over the codes and the dimensions $d$ for the ECCT is $\{4.58, 6.20, 8.43\}$ on the $\{4, 5, 6\}$ $E_b/N_0$ respectively, and $\{4.64, 6.37, 8.78\}$ for LS with one iteration.

| Method | ECCT | | | 1 It. Reg | | | 2 It. Reg | | | 3 It. Reg | | | 1 It. LS | | | 2 It. LS | | | 3 It. LS | | |
|---|---|---|---|---|---|---|---|---|---|---|---|---|---|---|---|---|---|---|---|---|---|
| | 4 | 5 | 6 | 4 | 5 | 6 | 4 | 5 | 6 | 4 | 5 | 6 | 4 | 5 | 6 | 4 | 5 | 6 | 4 | 5 | 6 |
| Polar(64,32) | 4.27 | 5.44 | 6.95 | 2.96 | 3.39 | 3.92 | 3.04 | 3.51 | 4.09 | 3.13 | 3.64 | 4.26 | 4.59 | 6.13 | 8.23 | 5.48 | 7.43 | 9.89 | 5.71 | 7.75 | 10.26 |
| | 4.57 | 5.86 | 7.50 | 2.96 | 3.39 | 3.92 | 3.05 | 3.52 | 4.09 | 3.14 | 3.65 | 4.26 | 4.87 | 6.51 | 8.65 | 5.84 | 7.86 | 10.23 | 6.05 | 8.14 | 10.59 |
| | 4.87 | 6.2 | 7.93 | 2.96 | 3.39 | 3.92 | 3.04 | 3.51 | 4.09 | 3.14 | 3.64 | 4.26 | 5.06 | 6.79 | 8.99 | 6.08 | 8.27 | 10.58 | 6.30 | 8.51 | 10.92 |
| Polar(64,48) | 4.92 | 6.46 | 8.41 | 3.78 | 4.41 | 5.18 | 3.92 | 4.63 | 5.46 | 4.07 | 4.85 | 5.75 | 5.03 | 6.80 | 9.13 | 5.42 | 7.42 | 9.92 | 5.52 | 7.58 | 10.15 |
| | 5.14 | 6.78 | 8.9 | 3.78 | 4.42 | 5.18 | 3.93 | 4.63 | 5.46 | 4.08 | 4.85 | 5.75 | 5.18 | 7.03 | 9.47 | 5.59 | 7.62 | 10.19 | 5.69 | 7.76 | 10.33 |
| | 5.36 | 7.12 | 9.39 | 3.78 | 4.42 | 5.18 | 3.93 | 4.63 | 5.46 | 4.09 | 4.86 | 5.75 | 5.24 | 7.15 | 9.59 | 5.65 | 7.74 | 10.22 | 5.74 | 7.89 | 10.43 |
| Polar(128,64) | 3.51 | 4.52 | 5.93 | 2.92 | 3.35 | 3.88 | 2.97 | 3.43 | 4.00 | 3.02 | 3.52 | 4.12 | 3.72 | 4.94 | 6.64 | 4.45 | 6.30 | 8.54 | 4.78 | 6.77 | 9.01 |
| | 3.83 | 5.16 | 7.04 | 2.93 | 3.36 | 3.88 | 2.98 | 3.44 | 4.00 | 3.04 | 3.52 | 4.12 | 3.91 | 5.22 | 6.97 | 4.85 | 6.82 | 9.06 | 5.27 | 7.33 | 9.47 |
| | 4.04 | 5.52 | 7.62 | 2.93 | 3.36 | 3.88 | 2.98 | 3.44 | 4.00 | 3.04 | 3.53 | 4.13 | 4.05 | 5.39 | 7.10 | 5.11 | 7.14 | 9.23 | 5.58 | 7.67 | 9.63 |
| Polar(128,86) | 4.30 | 5.58 | 7.34 | 3.49 | 4.05 | 4.74 | 3.57 | 4.18 | 4.93 | 3.66 | 4.32 | 5.13 | 4.53 | 5.97 | 7.92 | 5.21 | 7.18 | 9.64 | 5.42 | 7.46 | 9.98 |
| | 4.49 | 5.90 | 7.75 | 3.50 | 4.05 | 4.75 | 3.59 | 4.19 | 4.94 | 3.68 | 4.33 | 5.14 | 4.81 | 6.38 | 8.51 | 5.58 | 7.61 | 10.17 | 5.78 | 7.88 | 10.49 |
| | 4.75 | 6.25 | 8.29 | 3.50 | 4.06 | 4.75 | 3.59 | 4.19 | 4.94 | 3.69 | 4.33 | 5.14 | 5.04 | 6.73 | 9.02 | 5.87 | 8.03 | 10.71 | 6.06 | 8.29 | 11.04 |
| Polar(128,96) | 4.56 | 5.98 | 7.93 | 3.74 | 4.37 | 5.13 | 3.85 | 4.54 | 5.36 | 3.96 | 4.71 | 5.60 | 4.75 | 6.40 | 8.73 | 5.29 | 7.36 | 9.97 | 5.45 | 7.59 | 10.19 |
| | 4.69 | 6.20 | 8.30 | 3.75 | 4.38 | 5.13 | 3.86 | 4.54 | 5.36 | 3.97 | 4.72 | 5.61 | 4.94 | 6.78 | 9.34 | 5.59 | 7.86 | 10.71 | 5.77 | 8.12 | 11.00 |
| | 4.88 | 6.58 | 8.93 | 3.75 | 4.38 | 5.14 | 3.87 | 4.55 | 5.37 | 3.99 | 4.73 | 5.61 | 5.22 | 7.27 | 10.06 | 5.95 | 8.46 | 11.48 | 6.13 | 8.71 | 11.73 |
| LDPC(49,24) | 4.51 | 6.07 | 8.11 | 2.92 | 3.36 | 3.88 | 3.01 | 3.48 | 4.05 | 3.10 | 3.61 | 4.23 | 4.51 | 6.12 | 8.39 | 5.02 | 6.95 | 9.62 | 5.13 | 7.12 | 9.87 |
| | 4.58 | 6.18 | 8.46 | 2.92 | 3.36 | 3.88 | 3.01 | 3.48 | 4.05 | 3.10 | 3.61 | 4.23 | 4.54 | 6.13 | 8.48 | 5.07 | 6.92 | 9.74 | 5.18 | 7.09 | 10.02 |
| | 4.71 | 6.38 | 8.73 | 2.92 | 3.36 | 3.88 | 3.01 | 3.48 | 4.05 | 3.10 | 3.62 | 4.23 | 4.58 | 6.19 | 8.57 | 5.10 | 6.96 | 9.77 | 5.21 | 7.12 | 10.00 |
| LDPC(121,60) | 3.88 | 5.51 | 8.06 | 2.91 | 3.34 | 3.86 | 2.96 | 3.42 | 3.98 | 3.01 | 3.50 | 4.11 | 3.79 | 5.32 | 7.62 | 4.35 | 6.62 | 10.16 | 4.50 | 6.91 | 10.50 |
| | 3.89 | 5.55 | 8.16 | 2.91 | 3.34 | 3.86 | 2.96 | 3.42 | 3.98 | 3.01 | 3.50 | 4.11 | 3.79 | 5.33 | 7.64 | 4.36 | 6.62 | 10.16 | 4.52 | 6.91 | 10.48 |
| | 3.93 | 5.66 | 8.51 | 2.91 | 3.34 | 3.86 | 2.96 | 3.42 | 3.98 | 3.01 | 3.50 | 4.11 | 3.80 | 5.34 | 7.66 | 4.38 | 6.65 | 10.29 | 4.53 | 6.93 | 10.64 |
| LDPC(121,70) | 4.63 | 6.68 | 9.73 | 3.19 | 3.68 | 4.29 | 3.26 | 3.80 | 4.45 | 3.34 | 3.92 | 4.62 | 4.57 | 6.58 | 9.50 | 5.32 | 8.06 | 11.80 | 5.45 | 8.24 | 12.08 |
| | 4.64 | 6.71 | 9.77 | 3.19 | 3.68 | 4.29 | 3.27 | 3.80 | 4.45 | 3.34 | 3.92 | 4.62 | 4.60 | 6.63 | 9.64 | 5.37 | 8.15 | 12.13 | 5.50 | 8.39 | 12.48 |
| | 4.67 | 6.79 | 9.98 | 3.19 | 3.68 | 4.29 | 3.27 | 3.80 | 4.45 | 3.34 | 3.92 | 4.62 | 4.61 | 6.65 | 9.70 | 5.38 | 8.20 | 12.24 | 5.52 | 8.43 | 12.60 |
| LDPC(121,80) | 5.27 | 7.59 | 10.08 | 3.47 | 4.03 | 4.72 | 3.57 | 4.18 | 4.93 | 3.68 | 4.34 | 5.16 | 5.28 | 7.67 | 10.92 | 6.04 | 9.03 | 12.73 | 6.17 | 9.25 | 12.99 |
| | 5.29 | 7.63 | 10.90 | 3.47 | 4.03 | 4.72 | 3.57 | 4.18 | 4.93 | 3.68 | 4.34 | 5.16 | 5.29 | 7.65 | 11.00 | 6.06 | 9.05 | 12.87 | 6.19 | 9.24 | 13.14 |
| | 5.30 | 7.65 | 11.03 | 3.47 | 4.03 | 4.72 | 3.57 | 4.18 | 4.93 | 3.68 | 3.68 | 3.68 | 5.29 | 7.65 | 11.00 | 6.06 | 9.10 | 12.91 | 6.19 | 9.25 | 13.20 |
| MacKay(96,48) | 4.95 | 6.67 | 8.94 | 2.95 | 3.39 | 3.92 | 3.04 | 3.51 | 4.09 | 3.12 | 3.63 | 4.26 | 5.00 | 6.86 | 9.37 | 5.96 | 8.30 | 11.10 | 6.11 | 8.49 | 11.29 |
| | 5.04 | 6.80 | 9.23 | 2.95 | 3.39 | 3.92 | 3.04 | 3.51 | 4.09 | 3.13 | 3.63 | 4.26 | 5.05 | 6.95 | 9.52 | 6.02 | 8.42 | 11.41 | 6.18 | 8.64 | 11.59 |
| | 5.17 | 7.07 | 9.64 | 2.95 | 3.39 | 3.92 | 3.04 | 3.51 | 4.09 | 3.13 | 3.63 | 4.26 | 5.04 | 6.94 | 9.53 | 6.01 | 8.45 | 11.47 | 6.18 | 8.65 | 11.70 |
| CCSDS(128,64) | 4.35 | 6.01 | 8.30 | 2.93 | 3.37 | 3.90 | 3.00 | 3.47 | 4.04 | 3.06 | 3.57 | 4.18 | 4.33 | 6.00 | 8.44 | 5.40 | 7.95 | 11.52 | 5.68 | 8.34 | 11.95 |
| | 4.41 | 6.09 | 8.49 | 2.93 | 3.37 | 3.90 | 3.00 | 3.47 | 4.04 | 3.06 | 3.57 | 4.18 | 4.34 | 6.04 | 8.49 | 5.43 | 8.11 | 11.55 | 5.71 | 8.55 | 12.01 |
| | 4.59 | 6.42 | 9.02 | 2.93 | 3.37 | 3.90 | 3.00 | 3.47 | 4.04 | 3.07 | 3.57 | 4.18 | 4.35 | 6.08 | 8.51 | 5.44 | 8.09 | 11.65 | 5.72 | 8.48 | 12.13 |
| BCH(63,36) | 3.79 | 4.87 | 6.35 | 3.17 | 3.67 | 4.26 | 3.25 | 3.79 | 4.43 | 3.33 | 3.92 | 4.61 | 4.11 | 5.48 | 7.34 | 4.43 | 5.98 | 8.05 | 4.53 | 6.15 | 8.27 |
| | 4.05 | 5.28 | 7.01 | 3.17 | 3.67 | 4.26 | 3.25 | 3.79 | 4.44 | 3.33 | 3.92 | 4.62 | 4.21 | 5.63 | 7.52 | 4.52 | 6.12 | 8.22 | 4.64 | 6.31 | 8.49 |
| | 4.21 | 5.50 | 7.25 | 3.18 | 3.67 | 4.27 | 3.26 | 3.80 | 4.44 | 3.35 | 3.93 | 4.62 | 4.37 | 5.91 | 7.95 | 4.76 | 6.55 | 8.90 | 4.91 | 6.78 | 9.23 |
| BCH(63,45) | 4.47 | 5.88 | 7.81 | 3.65 | 4.26 | 4.98 | 3.76 | 4.43 | 5.22 | 3.88 | 4.62 | 5.48 | 4.62 | 6.25 | 8.57 | 4.96 | 6.85 | 9.50 | 5.06 | 7.04 | 9.76 |
| | 4.66 | 6.16 | 8.17 | 3.65 | 4.26 | 4.98 | 3.77 | 4.44 | 5.23 | 3.89 | 4.63 | 5.48 | 4.73 | 6.44 | 8.83 | 5.10 | 7.09 | 9.81 | 5.21 | 7.29 | 10.05 |
| | 4.79 | 6.39 | 8.49 | 3.65 | 4.26 | 4.98 | 3.77 | 4.44 | 5.23 | 3.90 | 4.63 | 5.48 | 4.79 | 6.52 | 8.95 | 5.17 | 7.18 | 9.93 | 5.30 | 7.39 | 10.21 |
| BCH(63,51) | 4.60 | 6.05 | 8.05 | 3.96 | 4.63 | 5.46 | 4.09 | 4.84 | 5.74 | 4.22 | 5.05 | 6.03 | 4.78 | 6.47 | 8.88 | 4.99 | 6.83 | 9.40 | 5.05 | 6.98 | 9.59 |
| | 4.78 | 6.34 | 8.49 | 3.96 | 4.63 | 5.46 | 4.10 | 4.85 | 5.74 | 4.23 | 5.06 | 6.03 | 4.85 | 6.62 | 9.08 | 5.08 | 6.99 | 9.74 | 5.16 | 7.12 | 9.96 |
| | 5.01 | 6.72 | 9.03 | 3.96 | 4.63 | 5.46 | 4.10 | 4.85 | 5.74 | 4.24 | 5.06 | 6.03 | 4.88 | 6.66 | 9.16 | 5.11 | 7.04 | 9.79 | 5.18 | 7.19 | 9.97 |

# E NUMBER OF DIFFUSION STEPS

Table 4 presents the convergence statistics (mean and standard deviation of the number of iterations) for the regular reverse diffusion and the line search procedure. Evidently, the line-search approach substantially reduces the number of steps, especially for low SNRs where the improvement can reach one order of magnitude.

Table 4: A comparison between the line search procedure and the regular reverse diffusion. The $\Delta$ column denotes the difference between the logarithm of Bit Error Rate (BER) for three normalized SNR values (i.e., $\Delta = -\log(\text{BER}_{LS}) + \log(\text{BER}_{Reg})$).
The other columns represent the mean and standard deviation of the number of iterations of the reverse process until convergence, i.e., convergence to zero syndrome.

| Method | $\Delta$ N=2 | | | $\Delta$ N=6 | | | #It. Reg. N=2 | | | #It Reg. N=6 | | | #It. LS N=2 | | | #It LS N=6 | | |
|---|---|---|---|---|---|---|---|---|---|---|---|---|---|---|---|---|---|---|
| | 4 | 5 | 6 | 4 | 5 | 6 | 4 | 5 | 6 | 4 | 5 | 6 | 4 | 5 | 6 | 4 | 5 | 6 |
| Polar(64,32) | -0.08 | -0.13 | -0.10 | 0.04 | 0.04 | 0.41 | 13.4±7.5 | 8.9±6.3 | 5.4±5.1 | 12.8±6.8 | 8.8±6.1 | 5.4±5.1 | 1.6±3.1 | 1.0±1.2 | 0.8±0.5 | 2.6±3.0 | 1.7±2.1 | 1.1±1.5 |
| | -0.14 | -0.07 | -0.30 | -0.04 | 0.21 | 0.89 | 13.2±7.2 | 8.9±6.2 | 5.4±5.1 | 12.7±6.7 | 8.8±6.0 | 5.4±5.1 | 1.4±2.3 | 1.0±0.9 | 0.8±0.5 | 1.1±0.8 | 0.9±0.4 | 0.8±0.4 |
| | -0.01 | -0.05 | 0.10 | -0.11 | 0.07 | 0.65 | 13.0±7.1 | 8.8±6.1 | 5.4±5.1 | 12.7±6.7 | 8.8±6.0 | 5.4±5.1 | 1.3±2.1 | 1.0±0.8 | 0.8±0.5 | 1.1±0.7 | 0.9±0.4 | 0.8±0.4 |
| Polar(64,48) | -0.02 | 0.11 | 0.16 | 0.01 | 0.06 | 0.47 | 5.9±4.9 | 3.3±3.8 | 1.5±2.7 | 5.7±4.6 | 3.2±3.8 | 1.5±2.7 | 1.4±2.3 | 0.7±1.0 | 0.4±0.5 | 1.2±1.7 | 0.7±0.9 | 0.4±0.6 |
| | 0.01 | 0.14 | 0.30 | -0.03 | 0.12 | 0.43 | 5.8±4.8 | 3.2±3.8 | 1.5±2.7 | 5.7±4.6 | 3.2±3.7 | 1.5±2.7 | 1.3±2.0 | 0.7±0.9 | 0.4±0.5 | 0.9±1.0 | 0.6±0.6 | 0.4±0.5 |
| | 0.00 | 0.16 | 0.40 | -0.01 | 0.07 | 0.71 | 5.8±4.7 | 3.2±3.8 | 1.5±2.7 | 5.7±4.6 | 3.2±3.7 | 1.5±2.7 | 1.2±1.8 | 0.7±0.8 | 0.4±0.5 | 1.0±1.0 | 0.6±0.6 | 0.4±0.5 |
| Polar(128,64) | -0.19 | -0.24 | -0.30 | -0.32 | -0.33 | -0.60 | 26.7±14.6 | 16.4±9.3 | 10.2±6.8 | 22.0±9.3 | 15.4±7.5 | 10.0±6.6 | 5.5±12.8 | 1.9±4.6 | 1.1±1.4 | 2.0±3.5 | 1.3±1.1 | 1.0±0.6 |
| | -0.31 | -0.61 | -0.40 | -0.26 | -0.30 | -0.10 | 24.4±12.2 | 15.8±8.2 | 10.1±6.6 | 21.6±8.5 | 15.4±7.4 | 10.1±6.6 | 3.5±8.6 | 1.5±2.9 | 1.1±0.9 | 1.8±2.0 | 1.3±0.9 | 1.0±0.5 |
| | -0.24 | -0.54 | -0.80 | -0.12 | -0.20 | 1.00 | 23.5±11.1 | 15.6±7.9 | 10.1±6.6 | 21.4±8.0 | 15.4±7.4 | 10.1±6.6 | 2.9±6.8 | 1.4±2.1 | 1.0±0.7 | 1.7±1.2 | 1.3±0.8 | 1.0±0.5 |
| Polar(128,86) | -0.10 | -0.11 | -0.20 | -0.20 | -0.13 | -0.21 | 16.5±10.9 | 9.1±7.3 | 4.7±4.9 | 13.4±7.5 | 8.3±5.8 | 4.6±4.6 | 4.0±9.1 | 1.4±3.4 | 0.8±1.0 | 2.9±4.4 | 1.6±2.1 | 1.0±1.2 |
| | -0.06 | -0.04 | 0.00 | -0.30 | -0.34 | -0.33 | 15.5±10.0 | 8.8±6.8 | 4.6±4.8 | 13.1±6.9 | 8.3±5.7 | 4.6±4.6 | 3.4±8.3 | 1.3±3.2 | 0.8±0.9 | 1.3±2.4 | 1.0±0.6 | 0.7±0.5 |
| | -0.09 | -0.06 | -0.10 | -0.32 | -0.21 | -0.40 | 14.7±9.2 | 8.6±6.4 | 4.6±4.7 | 13.0±6.6 | 8.3±5.7 | 4.6±4.6 | 2.7±6.8 | 1.2±2.3 | 0.8±0.7 | 1.2±1.9 | 0.9±0.5 | 0.7±0.5 |
| Polar(128,96) | -0.10 | -0.13 | -0.20 | -0.11 | 0.04 | -0.16 | 12.6±8.8 | 6.5±5.9 | 3.1±3.9 | 10.5±6.4 | 6.1±5.0 | 3.1±3.8 | 3.6±7.4 | 1.2±2.7 | 0.6±0.8 | 2.2±3.4 | 1.2±1.4 | 0.7±0.8 |
| | -0.09 | -0.10 | -0.10 | -0.19 | 0.13 | 0.14 | 11.8±8.0 | 6.3±5.5 | 3.1±3.8 | 10.32±6.11 | 6.1±4.9 | 3.1±3.8 | 2.7±6.0 | 1.1±2.0 | 0.6±0.6 | 1.2±2.0 | 0.9±0.5 | 0.6±0.5 |
| | -0.16 | -0.12 | -0.20 | -0.15 | 0.00 | -0.13 | 11.1±7.3 | 6.2±5.2 | 3.1±3.8 | 10.2±5.9 | 6.1±4.9 | 3.1±3.8 | 2.0±4.4 | 1.0±1.3 | 0.6±0.5 | 1.1±1.5 | 0.9±0.5 | 0.6±0.5 |
| LDPC(49,24) | 0.06 | 0.03 | 0.23 | -0.05 | -0.31 | -0.16 | 11.5±7.0 | 7.4±5.7 | 4.4±4.5 | 10.9±6.3 | 7.3±5.4 | 4.4±4.5 | 2.2±5.1 | 1.0±1.9 | 0.7±0.7 | 2.2±3.6 | 1.3±1.5 | 0.8±0.8 |
| | 0.05 | -0.06 | 0.20 | -0.13 | -0.12 | -0.20 | 11.4±7.0 | 7.4±5.7 | 4.4±4.5 | 10.9±6.3 | 7.3±5.4 | 4.4±4.5 | 2.1±4.9 | 1.0±1.9 | 0.7±0.6 | 1.5±3.5 | 0.9±1.1 | 0.7±0.5 |
| | 0.07 | -0.10 | 0.20 | -0.12 | -0.13 | -0.41 | 11.4±7.0 | 7.4±5.6 | 4.4±4.5 | 10.9±6.3 | 7.3±5.4 | 4.4±4.5 | 2.1±4.8 | 1.0±1.9 | 0.7±0.6 | 1.4±3.4 | 0.9±1.1 | 0.7±0.5 |
| LDPC(121,60) | -0.14 | -0.20 | -0.20 | -0.07 | -0.30 | 0.00 | 26.7±16.7 | 15.1±8.6 | 9.4±5.7 | 21.9±11.5 | 14.4±6.5 | 9.3±5.6 | 10.1±21.2 | 2.1±7.0 | 1.0±1.3 | 6.0±13.3 | 2.3±3.1 | 1.5±1.1 |
| | -0.08 | -0.17 | -0.30 | -0.11 | -0.09 | -0.44 | 26.6±16.5 | 15.1±8.5 | 9.4±5.7 | 21.7±11.2 | 14.4±6.5 | 9.3±5.6 | 10.1±21.2 | 2.1±7.0 | 1.0±1.3 | 4.4±13.2 | 1.3±2.6 | 1.0±0.4 |
| | -0.18 | -0.30 | -0.30 | -0.18 | -0.47 | -0.23 | 26.5±16.5 | 15.1±8.4 | 9.4±5.7 | 21.5±10.8 | 14.4±6.4 | 9.3±5.6 | 9.8±20.9 | 2.0±6.9 | 1.0±1.2 | 4.2±12.7 | 1.3±2.5 | 1.0±0.3 |
| LDPC(121,70) | -0.19 | -0.39 | -0.40 | -0.19 | -0.20 | -1.00 | 17.8±11.2 | 10.4±6.1 | 6.2±4.7 | 15.5±7.4 | 10.2±5.5 | 6.1±4.7 | 4.8±12.9 | 1.3±3.5 | 0.9±0.6 | 2.0±6.7 | 1.0±1.1 | 0.9±0.4 |
| | -0.16 | -0.29 | -0.20 | -0.16 | -0.44 | -0.17 | 17.6±10.9 | 10.4±6.0 | 6.2±4.7 | 15.5±7.3 | 10.2±5.5 | 6.1±4.7 | 4.5±12.3 | 1.2±3.2 | 0.9±0.6 | 1.9±6.2 | 1.0±0.9 | 0.9±0.4 |
| | -0.16 | -0.29 | -0.30 | -0.18 | 0.00 | -0.11 | 17.5±10.8 | 10.4±6.0 | 6.2±4.7 | 15.4±7.1 | 10.2±5.5 | 6.1±4.7 | 4.3±12.0 | 1.2±3.0 | 0.9±0.5 | 1.8±5.7 | 1.0±0.7 | 0.9±0.4 |
| LDPC(121,80) | -0.23 | -0.03 | -0.40 | -0.33 | -0.46 | -0.95 | 12.5±7.9 | 7.3±5.0 | 4.0±3.8 | 11.4±5.6 | 7.2±4.7 | 4.0±3.8 | 2.7±7.5 | 1.0±1.6 | 0.7±0.5 | 1.2±3.0 | 0.9±0.4 | 0.7±0.4 |
| | -0.13 | -0.10 | -0.30 | -0.15 | -0.42 | 0.81 | 12.5±7.9 | 7.3±4.9 | 4.0±3.8 | 11.4±5.8 | 7.2±4.7 | 4.0±3.8 | 2.7±7.7 | 1.0±1.7 | 0.7±0.5 | 1.4±3.7 | 0.9±0.6 | 0.7±0.4 |
| | -0.10 | -0.17 | -0.20 | -0.28 | -0.21 | -0.27 | 12.4±7.82 | 7.2±4.9 | 4.0±3.8 | 11.4±5.6 | 7.2±4.7 | 4.0±3.8 | 3.1±6.9 | 1.3±1.7 | 0.8±0.6 | 1.3±3.2 | 0.9±0.5 | 0.7±0.4 |
| MacKay(96,48) | -0.23 | -0.16 | -0.50 | -0.09 | 0.29 | 0.00 | 15.3±7.9 | 10.2±5.5 | 6.4±4.5 | 14.4±5.7 | 10.0±5.0 | 6.4±4.4 | 2.8±8.1 | 1.2±2.6 | 0.9±0.7 | 2.2±2.9 | 1.5±0.9 | 1.1±0.6 |
| | -0.20 | -0.14 | -0.30 | -0.17 | -0.21 | 0.00 | 15.3±7.8 | 10.2±5.4 | 6.4±4.5 | 14.3±5.7 | 10.0±5.0 | 6.4±4.4 | 2.6±7.8 | 1.2±2.5 | 0.9±0.7 | 1.6±2.1 | 1.0±0.5 | 0.9±0.3 |
| | -0.19 | -0.17 | -0.20 | -0.19 | -0.33 | -0.13 | 15.2±7.7 | 10.2±5.4 | 6.4±4.5 | 14.3±5.6 | 10.0±5.0 | 6.4±4.4 | 2.6±7.6 | 1.2±2.4 | 0.9±0.6 | 1.2±2.3 | 1.0±0.4 | 0.9±0.3 |
| CCSDS(128,64) | -0.19 | -0.58 | -0.40 | -0.31 | -0.36 | 0.52 | 20.8±11.4 | 13.1±6.3 | 8.4±4.9 | 18.2±6.6 | 12.8±5.4 | 8.4±4.9 | 4.7±13.4 | 1.4±4.1 | 1.0±0.8 | 1.7±4.7 | 1.1±0.7 | 1.0±0.3 |
| | -0.21 | -0.30 | -0.60 | -0.23 | -0.42 | 0.31 | 20.6±11.2 | 13.1±6.2 | 8.4±4.9 | 18.1±6.4 | 12.8±5.4 | 8.4±4.9 | 4.4±12.8 | 1.3±3.3 | 1.0±0.7 | 1.6±4.1 | 1.1±0.6 | 1.0±0.3 |
| | -0.27 | -0.42 | -0.40 | -0.26 | -0.11 | 0.37 | 20.6±11.0 | 13.1±6.2 | 8.4±4.9 | 18.1±6.3 | 12.8±5.4 | 8.4±4.9 | 4.2±12.3 | 1.3±3.4 | 1.0±0.6 | 1.3±3.7 | 1.1±0.5 | 1.0±0.3 |
| BCH(63,36) | -0.01 | 0.02 | -0.04 | 0.01 | 0.00 | 0.09 | 12.6±8.0 | 7.8±6.7 | 4.3±5.1 | 11.9±7.5 | 7.6±6.4 | 4.3±5.0 | 3.7±7.2 | 1.4±3.4 | 0.7±1.3 | 4.3±6.8 | 2.0±3.6 | 1.0±1.8 |
| | -0.01 | 0.02 | 0.13 | -0.11 | -0.05 | 0.03 | 12.4±7.9 | 7.7±6.6 | 4.3±5.1 | 11.8±7.4 | 7.5±6.3 | 4.3±5.0 | 3.3±6.8 | 1.3±3.1 | 0.7±1.1 | 2.5±5.6 | 1.1±2.4 | 0.7±0.8 |
| | 0.04 | 0.16 | 0.23 | -0.11 | -0.02 | 0.04 | 12.1±7.6 | 7.6±6.5 | 4.3±5.0 | 11.7±7.3 | 7.5±6.3 | 4.3±5.0 | 2.5±5.5 | 1.1±2.2 | 0.7±0.8 | 2.5±5.6 | 1.1±2.3 | 0.7±0.8 |
| BCH(63,45) | -0.05 | 0.00 | 0.11 | -0.10 | -0.05 | 0.37 | 7.2±5.7 | 4.1±4.5 | 2.0±3.2 | 6.9±5.3 | 4.0±4.4 | 1.9±3.2 | 2.1±3.9 | 0.9±1.6 | 0.4±0.6 | 1.5±3.0 | 0.7±1.1 | 0.4±0.5 |
| | 0.02 | 0.15 | 0.10 | -0.14 | 0.08 | 0.41 | 7.1±5.6 | 4.0±4.5 | 2.0±3.2 | 6.8±5.2 | 3.9±4.3 | 2.0±3.2 | 1.9±3.6 | 0.8±1.4 | 0.4±0.6 | 1.4±2.6 | 0.7±0.9 | 0.4±0.5 |
| | 0.03 | 0.18 | 0.20 | -0.13 | -0.06 | 0.73 | 7.1±5.6 | 4.0±4.5 | 2.0±3.2 | 6.8±5.3 | 3.9±4.3 | 2.0±3.2 | 1.8±3.4 | 0.8±1.3 | 0.4±0.6 | 1.3±2.5 | 0.7±1.0 | 0.4±0.5 |
| BCH(63,51) | 0.04 | 0.28 | 0.94 | 0.09 | 0.42 | 1.16 | 4.8±4.1 | 2.6±3.4 | 1.2±2.4 | 4.7±4.0 | 2.6±3.4 | 1.2±2.4 | 1.8±2.9 | 0.7±1.3 | 0.3±0.6 | 1.6±2.6 | 0.7±1.2 | 0.3±0.6 |
| | 0.06 | 0.31 | 1.13 | 0.06 | 0.28 | 1.19 | 4.8±4.1 | 2.6±3.4 | 1.2±2.4 | 4.7±4.0 | 2.6±3.4 | 1.2±2.4 | 1.6±2.7 | 0.7±1.2 | 0.3±0.5 | 1.3±2.1 | 0.6±1.0 | 0.3±0.5 |
| | 0.04 | 0.34 | 1.02 | 0.02 | 0.34 | 1.04 | 4.8±4.1 | 2.6±3.4 | 1.2±2.4 | 4.7±4.0 | 2.6±3.4 | 1.2±2.3 | 1.6±2.6 | 0.7±1.1 | 0.3±0.5 | 1.4±2.3 | 0.6±1.0 | 0.3±0.5 |

# F DENOISING DIFFUSION ERROR CORRECTION CODE TRANSFORMER

## F.1 PARITY-CHECK CONDITIONING ARCHITECTURE

The DDECCT architecture is depicted in Figure 9. In order to condition the network on the number of parity errors $e_t \in \{0, \ldots, n-k\}$, we employ a $d$ dimensional one hot encoding multiplied via Hadamard product with the initial elements' embedding of the ECCT. Denoting the ECCT's embedding of the $i$ element as $\phi_i$, the new embedding is defined as $\tilde{\phi}_i = \phi_i \odot \psi(e_t), \forall i$, where $\psi$ denotes the $n-k$ one hot embedding.

## F.2 DDECCT COMPUTATIONAL COMPLEXITY

The computational overhead consists of the conditioning, which is a negligible Hadamard product of the initial ECCT embedding with the one-hot encoding of the number of parity-check errors, the parallel line-search procedure (which scales linearly with the density of the code, and most codes are sparse), and the number of reverse diffusion iterations, which is reduced to extremely few iterations by the line-search framework.

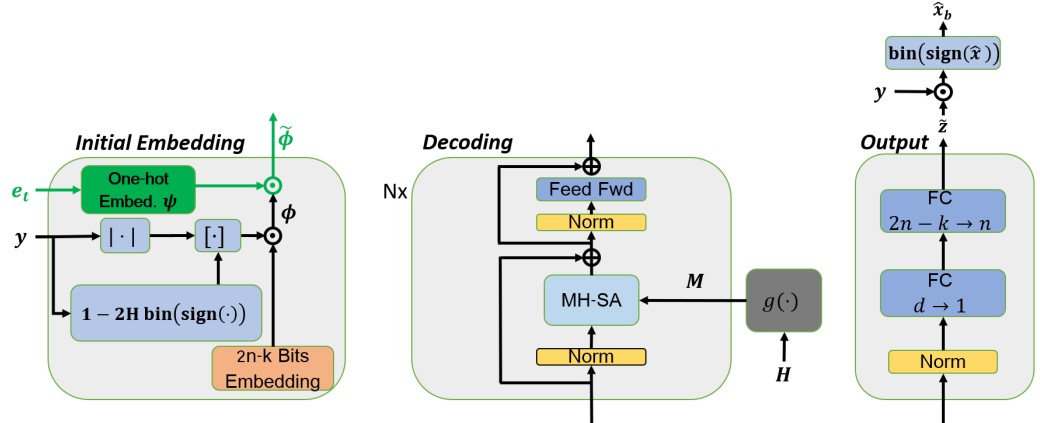

Figure 9: Illustration of the proposed DDECCT architecture. The main difference from ECCT is the green component in the Initial Embedding module.

Therefore, the large improvement of DDECCT over ECCT is obtained while employing a modest increase in computational complexity. Most importantly, the space complexity, determining the capacity of the network, is extremely reduced with the DDECC framework, since even shallow DDECCT architectures outperform deep ECCT models.

It should also be mentioned that as described in the introduction, the iterative DDECC framework supports a differential treatment of the samples based on their level of corruption, what is not possible with ECCT.

## G  BETA SCHEDULING

The choice of the scheduling range, i.e. $t \in \{0, 1, \ldots, T\}$ is explained as follows. In contrast to classical denoising diffusion models used as generative models, the conditioning in our model is performed over the number of parity-check errors in order to obtain an estimate of the proximity to the solution. Thus, $T$ is now defined as the maximum possible number of parity-check errors, i.e. $n - k$.

The values of the beta schedule have been set empirically such that the cumulative sum of the diffusion noise roughly corresponds to the average training noise statistics, i.e., $\sqrt{\bar{\beta}_t} \approx \sigma$ such that $y = x + \sigma z, z \sim \mathcal{N}(0, I)$, as described in Eq. 8.

In addition to the empirical support, constant scheduling has been chosen to induce a uniform treatment of the noise space. Finally, the full method uses the line-search procedure, reducing the need to precisely define or tune the scheduler.

## H  COMPARISON WITH SUCCESSIVE CANCELLATION LIST (SCL) POLAR DECODER

Table 5 compares the performance of ECCT and DDECCT to the SOTA SCL Polar decoder Tal & Vardy (2015) for several Polar Codes. The SCL decoder has a time and space complexity of $\mathcal{O}(LN \log N)$ and $\mathcal{O}(LN)$, respectively.

We tested the SCL algorithm for $L = \{1, 4\}$ using the AFF3CT software (Cassagne et al., 2019). Note that with this software package, we cannot ensure an exact comparison to our settings and even with regard to the polar code used.

Increasing the capacity of the network, especially with more layers, is expected to lead to better results as demonstrated for LDPC codes. Similarly, SCL with bigger lists would obtain improved accuracy. We present in Figure 10 the performance of different architecture size. We can observe larger architecture is able to close the gap with the SOTA on Polar codes.

Table 5: A comparison of the negative natural logarithm of Bit Error Rate (BER) for three normalized SNR values (4,5,6) between the proposed method with $N = 6, d = 128$, the ECCT and the SOTA SC-L algorithm. Higher is better.

| Method | SC-$L = 1$ | | | SC-$L = 4$ | | | ECCT | | | DDECCT | | |
|---|---|---|---|---|---|---|---|---|---|---|---|---|
| | 4 | 5 | 6 | 4 | 5 | 6 | 4 | 5 | 6 | 4 | 5 | 6 |
| Polar(64,32) | 7.30 | 9.67 | 13.18 | 8.11 | 10.70 | 14.04 | 6.62 | 8.89 | 12.02 | 6.93 | 9.51 | 12.79 |
| Polar(64,48) | 6.17 | 8.41 | 10.97 | 6.63 | 8.63 | 11.24 | 6.19 | 8.28 | 11.05 | 6.00 | 8.24 | 10.98 |
| Polar(128,64) | 8.37 | 11.69 | 13.70 | 9.60 | 13.16 | 17.42 | 5.92 | 8.64 | 12.18 | 9.11 | 12.90 | 16.30 |
| Polar(128,86) | 7.54 | 10.74 | 15.14 | 9.26 | 13.04 | 17.13 | 6.31 | 9.01 | 12.45 | 7.60 | 10.81 | 15.17 |
| Polar(128,96) | 6.74 | 9.53 | 13.53 | 8.02 | 11.60 | 18.16 | 6.31 | 9.12 | 12.47 | 7.16 | 10.3 | 13.19 |

We believe the high number of degrees of freedom in our model coupled with careful training and hyper-parameter tuning should fill the remaining gap. Most importantly, we believe that permutations of the parity check matrix may have a large impact on the performance as described in many previous neural decoding works (e.g. (Bennatan et al., 2018; Raviv et al., 2020)).

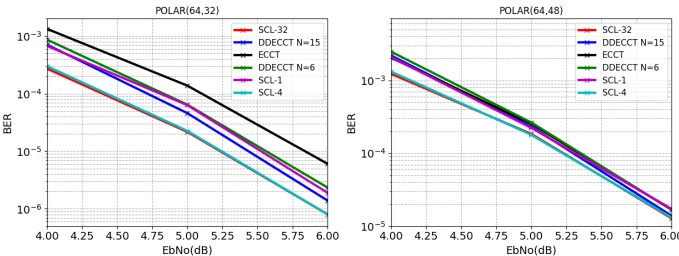

Figure 10: BER plots comparing ECCT, DDECCT and the SCL algorithm for various $Eb/N_0$ and $N$ values.

## I   RESULTS OVER MULTIPLE SNR VALUES

Figure 11 depicts BER plots comparing the performance of ECCT and DDECCT for several codes.

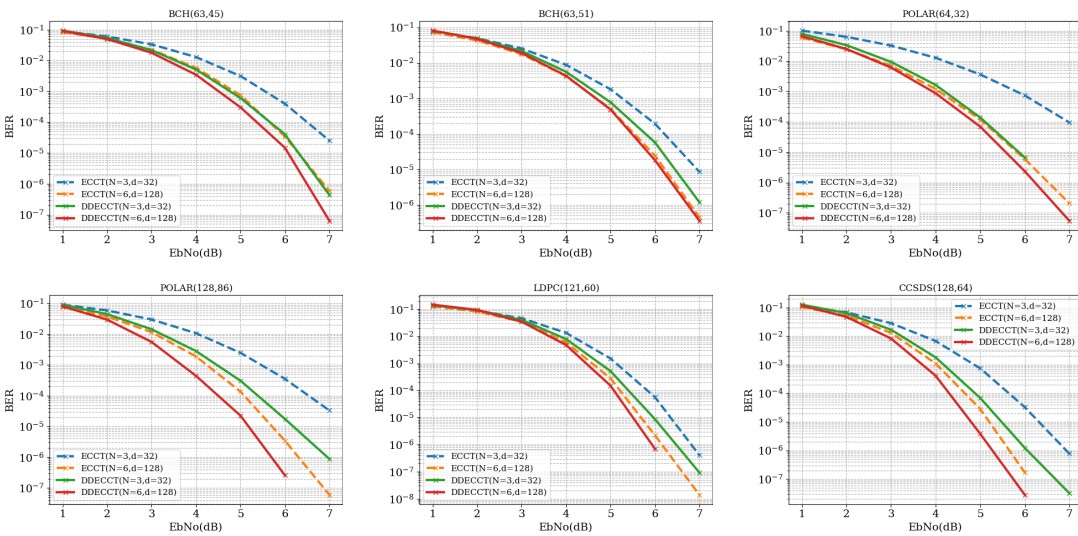

Figure 11: BER plots comparing ECCT and the proposed DDECCT for various $Eb/N_0$ values.

