# OpenReview forum: "Denoising Diffusion Error Correction Codes"
_ICLR.cc/2023/Conference — ICLR 2023 notable top 25%_

### Official Review · Reviewer_U25g · 2022-10-17

**Confidence:** 4
**Clarity, Quality, Novelty And Reproducibility:** The paper is overall clear and well w…
**Correctness:** 4
**Technical Novelty And Significance:** 3
**Empirical Novelty And Significance:** 3
**Recommendation:** 8

**Strength And Weaknesses:**

Strengths:

(1) The idea of using diffusion processes in the context of error correction is, at the best of my knowledge, new. Compared to existing neural decoders which often do not scale well to larger block lengths, this novel framework has the potential to scale to codes with block lengths of a few hundreds of bits.

(2) Clear gains in a number of different settings with respect to ECCT, i.e., the transformer-based approach of Choukron & Wolf, (2022).


Weaknesses:

(1) My main concern is that the comparison with 'classical' (i.e., non neural-network-based) decoding algorithms is not sufficiently thorough. It could be that DDECCT does better than ECCT, but ECCT is not quite SOTA, in the sense that there is a 'classical' algorithm that performs much better with roughly the same complexity. More specifically, the standard decoder of polar codes is *not* the BP decoder taken into account here, but rather the successive cancellation list (SCL) decoder. Is DDECCT able to match SCL performance? Let me highlight that, at the lengths considered here (64 in Figure 4, but also 512 in Figure 5), SCL reaches the optimal ML performance with a relatively small list size.

(2) In page 8, the authors give an expression of the overhead of the proposed method over ECCT. However, it is not clear what the overhead is in terms of running time. Suppose that one wishes to compare ECCT with the method proposed here, keeping the complexity fixed. This means that ECCT could use a larger N or d than DDECCT (provided that the complexity of the two procedures is the same). Would DDECCT still compare favourably?

(3) Comparing the performance of different decoding algorithms in the tabular form of Table 1 is not really standard in wireless communications. In particular, what really matters in practice is the SNR needed to reach a certain target BER. Target BERs typical in wireless communications are e.g. in the range $10^{-3}-10^{-5}$. This difference in reporting the results can be quite substantial: if the error curves are very steep, the improvement in the error probability at a fixed SNR could seem very large, while the improvement in the SNR needed to reach the target BER may be limited. I strongly suggest that the authors provide a plot of the error probability as a function of the SNR in addition to Table 1, so that the improvement with respect to the baselines is more clear. As a final (minor) note, I would suggest that the authors put in bold the number corresponding to the best performance in Table 1 to improve readability.

**Summary Of The Paper:**

This paper proposes to employ denoising diffusion models to decode error correcting codes (ECCs). In particular, the authors introduce a diffusion process that models well the decoding of an ECC, and the denoising is performed by conditioning on the number of errors in the parity check matrix. In order to choose the step size of the reverse diffusion, a line search is adopted. The authors compare their results to the transformer-based decoder recently proposed by Choukron & Wolf, (2022). Some comparisons with 'classical' decoders (BP) are also provided.

**Summary Of The Review:**

Overall, the paper provides a new framework based on denoising diffusion models, which results in improvements over the existing work by Choukron & Wolf, (2022). My borderline initial opinion is due to the weaknesses pointed out above concerning the experimental results. If such points are properly addressed during the rebuttal, I can raise my score.

---

My concerns have been addressed and I have increased my score.

---

> ### Author Response · Authors · 2022-11-16
> **Authors response**
>
> Thank you for the supportive and comprehensive review.
>
> ## Comparison to classical decoders
>
> The main focus of this work is to improve the speed and decoding capabilities of existing neural decoders on a broad range of code families. We chose the ECCT since it is the current SOTA neural decoder. The results indicate that existing neural decoders (and our method) outperform the best classical decoders on the code family for which each classical decoder was designed (e.g. LDPC and BP).
>
> In Appendix H, we now add, besides BP, the performance of SCL for polar codes. Relatively shallow ECCTs can compete and sometimes even surpass the SC-L for some of the codes and SNRs. DDECCT outperforms ECCT and SC-L by a large margin. Increasing the capacity of the network, which currently has only six layers, may further improve the performance.
>
> SC-L complexity is $\mathcal{O}(Ln \log n)$ while the ECCT/DDECCT scales linearly with the code length. We could not run higher order of the SC-L because of the high complexity and non-parallel application of the classical SC-L algorithm.
>
> A rigorous comparison should take into account the exact complexity analysis as well as the potential acceleration of the ECCT as suggested in Section 6.3. of the ECCT paper. For example, a low-rank approximation, e.g. Linformer [32], would transform the quadratic complexity in $d$ to linear, which could make ECCT extremely competitive on the algorithmic complexity level as well.
>
>
> ## The computational overhead of DDECCT over ECCT
>
> The computational overhead consists of the conditioning, which is a negligible Hadamard product of the initial ECCT embedding with the one-hot encoding of the number of parity-check errors, the *parallel* line-search procedure (equivalent to the constant complexity computation of the syndrome), and the number of reverse diffusion iterations, which is reduced to extremely few iterations by the line-search framework.
>
> Therefore, the large improvement of DDECCT over ECCT is obtained while employing a modest increase in computational complexity.  Most importantly, the space complexity, determining the capacity of the network, is extremely reduced with the DDECC framework, since even shallow DDECCT architectures outperform deep ECCT models.
>
> It should also be mentioned that as described in the introduction, the DDECC framework supports a differential treatment of the samples based on the level of corruption. This is not possible with ECCT.
>
> These clarifications are now part of the manuscript, see Appendix F of the revised version.
>
>
> ## Graph-like experiments
> We followed the performance presentation of previous works (e.g. [1,2]) in a table form, allowing us the display a greater number of codes. We now provide results in the requested graph format for several codes in Appendix I.
> We also put in bold the best performance of Table 1, as suggested by the reviewer.
>
> [1] Hyper-graph-network decoders for block codes, by Nachmani and Wolf.
>
> [2]  perm2vec: Graph permutation selection for decoding of error correction codes using self-attention, by Raviv et al.

---

> > ### Comment · Reviewer_U25g · 2022-11-18
> > **Still concerned about the comparison with SCL decoding of polar codes**
> >
> > I would like to thank the authors for the thoughtful response and revision. However, I am puzzled by the fact that, according to Table 5 in Appendix H, having L=4 does not seem to improve performance w.r.t. L=1. This is clearly in contrast with the results of (Tal & Vardy, 2015), which display a significant performance improvement of list decoding (L>1) over standard successive cancellation decoding (L=1). This is a red flag on the validity of the new simulation results provided by the authors.
> >
> > On a secondary note, I find the following quite an overstatement: "The corresponding DDECCT outperforms the other methods by very large margins". For Polar(64, 48), SCL with L=1 still provides the best performance for all the three SNRs taken into account.

---

> > > ### Author Response · Authors · 2022-11-19
> > > **Authors response**
> > >
> > > We thank the reviewer for the clear feedback.
> > >
> > > ## Lack of improvement with L>1 for the SCL decoder.
> > > We had some challenges finding an open-source python library that supports efficient integration and equivalent comparison between the SCL decoding and our settings.
> > > Out of not too many options for python libraries (https://github.com/topics/polar-codes?l=python), we used the following library: https://github.com/fr0mhell/python-polar-coding ; https://github.com/fr0mhell/python-polar-coding/tree/master/python_polar_coding/polar_codes/sc_list ).
> > >
> > > We acknowledge the fact increasing the number of decoding paths seems to improve the performance only slightly. We assumed that for short codes, SCL already (almost) reaches ML decoding performance while for larger codes it would require higher Ls.
> > >
> > > We will try to run experiments with alternative SCL libraries. Every suggestion of open-source software allowing fair comparison is greatly welcomed.
> > >
> > > ## Overstatement.
> > > The text will be modified as follows:
> > > “The corresponding DDECCT outperforms the other methods by large margins for larger codes ($n=128$).”

---

> > > > ### Comment · Reviewer_U25g · 2022-11-21
> > > > **Sanity check**
> > > >
> > > > One sanity that should be performed is to recover the numerical results in (Tal & Vardy, 2015) (or in any of the many follow-up works that use SCL decoding). There, the authors consider much longer codes (2k block length), where the advantage of the list is significant.

---

> > > ### Author Response · Authors · 2022-11-21
> > > **Authors Response**
> > >
> > > Following the reviewer's recommendation, we performed SCL experiments on Polar codes with the aff3ct software https://github.com/aff3ct/aff3ct/tree/development, which seems to be more mature than the SCL package we previously used.
> > >
> > > The new results are provided in the table below and will be added to the manuscript. As the reviewer indicated, SCL reaches almost ML decoding even for small $L$s (e.g., $L=4$) for short codes, while the contribution of larger $L$s is substantial for larger codes.
> > > In terms of BER, for short codes, SCL with $L=32$ is better than the *shallow* DDECCT by a factor of ~1-3 (the table displays logarithmic scale), while for larger codes the factor ranges from ~1 to 10 (120 for POLAR(128,96) at Eb/N0=6).
> > >
> > > | Method             |  	|  L=1  |   	|  	|  L=4  |   	|      |  L=32 |   	|      |  Our  |       |
> > > |--------------------|:----:|:-----:|:-----:|:----:|:-----:|:-----:|:----:|:-----:|:-----:|:----:|:-----:|:-----:|
> > > | **Eb/N0**          |   4  |   5   |   6   |   4  |   5   |   6   |   4  |   5   |   6   |   4  |   5   |   6   |
> > > | **POLAR (64,32)**  | 7.30 |  9.67 | 13.18 | 8.11 | 10.70 | 14.04 | 8.21 | 10.74 | 14.04 | 6.93 |  9.51 | 12.79 |
> > > | **POLAR (64,48)**  | 6.17 |  8.41 | 10.97 | 6.63 |  8.63 | 11.24 | 6.70 |  8.60 | 11.27 | 6.00 |  8.24 | 10.98 |
> > > | **POLAR (128,64)** | 8.37 | 11.69 | 13.70 | 9.60 | 13.16 | 17.42 | 9.60 | 13.16 | 17.42 | 9.11 | 12.90 | 16.30 |
> > > | **POLAR (128,86)** | 7.54 | 10.74 | 15.14 | 9.26 | 13.04 | 17.13 | 9.36 | 13.07 | 17.14 | 7.60 | 10.81 | 15.17 |
> > > | **POLAR (128,96)** | 6.74 | 9.53  | 13.53 | 8.02 | 11.60 | 18.16 | 8.24 | 11.61 | 18.01 | 7.16 | 10.30 | 13.19 |
> > >
> > > Note that with this software package, one cannot ensure an exact comparison to our settings and even with regard to the polar code used. Permutation of the code can have a significant impact on neural decoders in general and on the (DD)ECCT in particular (see [1,2,3]).
> > >
> > > Now that we are challenged by a better decoder, we will try to use *deeper* DDEECCT networks for polar codes and report the results. We expect the performance gap to drop or even invert.
> > >
> > > It is worth reiterating that in a wide variety of codes and multiple noise settings, we were able to demonstrate a clear superiority over every existing neural decoder (with no exception).  This has been achieved with rather shallow networks and with a small training budget (i.e., small capacity models, no hyper-parameters search, short training) since such optimizations, albeit important, were not necessary.
> > >
> > > [1] Near Maximum Likelihood Decoding with Deep Learning, by Nachmani et al.
> > >
> > > [2] Deep learning for decoding of linear codes-a syndrome-based approach, by Bennatan et al.
> > >
> > > [3] perm2vec: Graph permutation selection for decoding of error correction codes using self-attention, by Raviv et al.

---

> > > > ### Comment · Reviewer_U25g · 2022-11-22
> > > > **More reasonable results**
> > > >
> > > > Thank you for the additional comparison, the new numbers you reported look credible and well in line with the existing literature. I look forward to seeing if deeper DDEECCT can improve over list decoding. This would indeed be great since, as you mentioned, SCL essentially achieves optimal ML performance with rather small values of the list size.
> > > >
> > > > Let me finally mention that showing results comparable to SCL (which is in the 5G standard) would be pretty impressive.

---

> > > > > ### Author Response · Authors · 2022-12-07
> > > > > **As promised: an update regarding polar codes**
> > > > >
> > > > > Following a comment by reviewer U25g, we have changed the software packages used for evaluating SCL and, as a result, noticed that DDECCT’s performance is lower than the SCL method, which is known to achieve the maximum likelihood bound on short-length codes.
> > > > >
> > > > > As promised in our comment from November 21, we have tried larger architectures. Our results for the (64,32) and (64,48) Polar codes, respectively, are shown in:
> > > > >
> > > > > https://ibb.co/YDjsH8B
> > > > >
> > > > > https://ibb.co/hd3qVQx
> > > > >
> > > > > Evidently, increasing the depth of the DDECCT allows us to be comparable to large list-length SCL (i.e. maximum likelihood) performance on the shorter-length POLAR codes.
> > > > >
> > > > > We believe the high number of degrees of freedom in our model coupled with careful training and hyper-parameter tuning should fill the remaining gap. Most importantly, we believe that permutations of the parity check matrix may have a large impact on the performance as described in many previous neural decoding works (e.g. [1,2,3]).
> > > > >
> > > > > Needless to say, DDECCT obtains a sizable gap in performance over all other deep learning methods for Polar codes and is the state of the art for all of the many non-Polar codes that we tested.
> > > > >
> > > > > Thank you for helping us improve our work.
> > > > >
> > > > >
> > > > > [1] Near Maximum Likelihood Decoding with Deep Learning, by Nachmani et al.
> > > > >
> > > > > [2] Deep learning for decoding of linear codes-a syndrome-based approach, by Bennatan et al.
> > > > >
> > > > > [3] perm2vec: Graph permutation selection for decoding of error correction codes using self-attention, by Raviv et al.

---

> > > > > > ### Comment · Reviewer_U25g · 2022-12-09
> > > > > > **Thank you for the follow up**
> > > > > >
> > > > > > Thank you for carrying out the promised additional simulations. The new results are interesting and I strongly suggest that you include them in the revision of the paper. At the same time, I also suggest the authors to be cautious in the way they present their (good) results: there is still a clear gap between their proposed neural decoder and the state-of-the-art SCL. Closing this gap is an exciting avenue for future research and, in my opinion, would allow for neural decoders to actually be considered for standardization. Thus, the authors should simply admit this gap and mention a few ways in which the gap can be reduced (as done by the authors in the comment).
> > > > > >
> > > > > > All that being said, I am satisfied with the response and will increase my rating accordingly.

---

### Official Review · Reviewer_YkMp · 2022-10-27

**Confidence:** 2
**Correctness:** 3
**Technical Novelty And Significance:** 3
**Empirical Novelty And Significance:** 3
**Recommendation:** 6

**Clarity, Quality, Novelty And Reproducibility:**

Clarity: The presentation is not clear. I think the derivation/explanation of the algorithms can be much simpler or convincing. For example, the current explanation does not provide sufficient reasoning as to why the multiplicative noise is treated and inferred instead of the additive noise that is directly affecting the codeword; hence the reader may wonder what happens if inferring the additive noise directly. Also, there are typos and misleading expressions at some important points in the overall manuscript. The following is the list of them I have realized:
******************************************************
Page 3, below eq. (2): The equation \hat{x}=y\cdot \epsilon_{\theta} should be \hat{x}=y\cdot (\epsilon_{\theta})^{-1} by the definition of the multiplicative noise. I know it finally becomes multiplicative sign noise so (\epsilon_{\theta})^{-1}=\epsilon_{\theta}, but at this point it is not explained that only the sign of the multiplicative noise is important.

Page 3, above eq. (5): The sentence ``a model \epsilon_{\theta}(x_t,t) that predicts the additive noise \epsilon'' can be misleading since later the symbol \epsilon_{\theta} is used as an estimate of the ``multiplicative noise''. One should use different notations between these two things.

Page 4, eq. (10): q(x_t|x_{t-1}) should not be q(x_{t-1}|x_t)? The same is true for q(x_t|x_{t-1}) above eq. (10)

Page 4, eq. (12): Why use ``log'' with two arguments to denote binary cross-entropy? I think it is better to use another notation. The same is true for eq. (13)

Page 6, Alg. 2: The meaning of the function ``e'' should be explained.

Page 6, Alg. 2: How to compute \lambda is not well explained. How to relate y and x_t in eq. (16)? I think it is better to define a function to compute \lambda and use it for explanation.

Page 6, in Sec. 4.5: Although the variance schedule is fixed to \beta_t=0.01, the range of t is not explained.

Page 9, in Sec. 5.2: Figure 2 > Figure 6
******************************************************
Overall, I think the level of clarity is low.

Quality: Although the presentation is not good, the idea itself is interesting. The implementation seems to work well and the proposed method largely outperforms the other methods. These points make the quality of the papers high.

Novelty: The base idea using the diffusion model in decoding is novel. The additional ideas for implementation are also original. The level of novelty is thus high.

Reproducibility: Although I did not check the code, the algorithms and the learning details are explained and thus one can reproduce the result in principle. Hence there is no serious problem in reproducibility.


**Strength And Weaknesses:**

Strengths:
- A novel idea about how to use the diffusion model in decoding.

- The high decoding performance of the proposed method compared to other methods.

Weaknesses:
- The ideas for implementation, such as learning the binarized multiplicative noise instead of the additive one and learning the noise process conditioned by the number of parity check errors, have no theoretical reasoning.

- There seems to be no guideline on how to choose the variance schedule {\beta_t}_{t=0}^{T}.


**Summary Of The Paper:**

This work proposes a new decoding framework in algebraic block codes by employing a denoising diffusion model.
The idea is to learn the process of the codeword corruption using the diffusion model and to detect the noise contribution in an output y by computing the reverse process of the corruption using the learned model. To implement this idea, the authors make some approximations in the reverse process and propose some additional ideas, such as learning the binarized multiplicative noise instead of the additive one and learning the noise process conditioned by the number of parity check errors. The experimental results of the proposed method significantly outperform the other comparative methods, such as the transformer-based framework and the standard decoding scheme based on Belief propagation in terms of the bit error rate (BER), demonstrating the superiority of the proposed method.


**Summary Of The Review:**

The core idea of the paper is novel and interesting, and the experimental result of the proposed method is good. Hence, I basically vote for accepting the paper, though the presentation should be improved.

---

> ### Author Response · Authors · 2022-11-16
> **Authors response**
>
> We thank the reviewer for the thorough and comprehensive review.
>
> ## Theoretical reasoning of the binarized multiplicative noise
>
> The multiplicative noise is an equivalent statistical model to the true physical additive one since $y=x_{s}+z=x_{s}*\tilde{z} \Rightarrow \tilde{z}=yx_{s}$. It is chosen in order to allow the magnitude to be *independent* of the transmitted codeword $x$, allowing efficient training without the need to sample the code space and thus avoiding overfitting in model-free architectures as described in the introduction and the related work section.
> This setting is adopted from *Bennatan et al.* 2018 and results directly from the proof of Lemma 1 in *“The Capacity of Low-Density Parity-Check Codes Under Message-Passing Decoding”* by  *Richardson and Urbanke*.
>
> It can be demonstrated as follows:
> $P(|y| |x)=P(|x_{s}\tilde{z}||x)=P(|x_{s}||\tilde{z}| |x)=P(|\tilde{z}| |x)=P(|\tilde{z}|)$.
> The syndrome is known as independent of the transmitted codeword since the code defines the kernel of the parity-check matrix mapping, i.e. $H(x+e)=Hx+He=He$.
> The full motivation of this approach for neural decoding and the proof of optimality can be found in Bennatan et al. 2018 Sections IV.B. and IV.C.
>
> The multiplicative noise setting is more a constraint of the chosen neural decoder (i.e. ECCT) rather than the denoising diffusion framework but is necessary in order to eliminate the risk of overfitting. Thus, the DDECC framework conciliates between the ECC overfitting constraint (i.e. the multiplicative noise) and the DDPM legacy formulation (i.e. the additive noise).
> This reasoning is made clearer in Section 3 of the revised manuscript.
>
> ## Choice of the beta schedule and range
>
> The choice of the scheduling range, i.e. $t\in \{0,1,\dots ,T\}$ is given in Section 4.3 below equation 13.
> In contrast to classical denoising diffusion models used as generative models, the conditioning in DDECC is performed over the number of parity-check errors in order to obtain an estimate of the proximity to the solution. Thus, $T$ is now defined as the maximum possible number of parity-check errors, i.e. $n-k$.
>
> The values of the beta schedule have been set empirically such that the cumulative sum of the diffusion noise roughly corresponds to the average training noise statistics, i.e., $\sqrt{\bar{\beta_t}} \approx \sigma$ such that $y=x_{s}+\sigma z$, $z \sim \mathcal{N}(0,I)$, as described in Eq. (8).
>
> Besides the empirical support, constant scheduling has been chosen to introduce a uniform treatment of the noise space.
>
> Finally, the full method uses the line-search procedure, which reduces the need to precisely define or tune the scheduler.
> We added these explanations to the new Appendix G of the revised manuscript.
>
> ## Typos and misleading expressions
> We thank the reviewer for these important remarks. They have been corrected in the revised manuscript.
>
> **Page 3, below eq. (2):**
> We now formulate the prediction as the hard decoding task.
>
> **Page 3, above eq. (5):**
> We modified the misleading notation, it is now referred to as $\epsilon^{DDPM}_{\theta}$.
>
> **Page 4, eq. (10):**
> We corrected the confused posterior notation.
>
> **Page 4, eq. (12):**
> We removed this abuse of notation in the revised manuscript.
>
> **Page 6, Alg. 2:**
> The function $e$ is defined in section 4.3, above eq. (13).
>
> **Page 6, Alg. 2:**
> $\lambda$ is defined as the solution of the optimization problem that is defined in eq. (16).
> The proposed grid-search procedure is sub-optimal and is stated as part of the implementation rather than the algorithm, where other line-search procedures can be suggested.
> Also, eq (16) is not related to $y$ but to the current iterate $x_{t}$, except that the initial iterate corresponds to the transmitted signal, i.e. $x_{t}=y$.
>
> **Page 6, in Sec. 4.5:**
> We added explanations regarding the scheduler in the new Appendix G.
>
> **Page 9, in Sec. 5.2:**
> The labeling error has been fixed.

---

> > ### Comment · Reviewer_YkMp · 2022-11-23
> > **thank you for the explanations**
> >
> > Thank you for the detailed explanations that are satisfactory to me. However, since I am not very familiar with the field, it isn't easy to judge the impact of the research, so I would like to keep the current score which is in the positive side from the beginning.

---

### Official Review · Reviewer_8AQR · 2022-10-31

**Confidence:** 3
**Correctness:** 3
**Technical Novelty And Significance:** 3
**Empirical Novelty And Significance:** 3
**Recommendation:** 8

**Clarity, Quality, Novelty And Reproducibility:**

Most part of the paper is easy to read and well organized. From the attached codes, one interested in reproducing the results may be able to reproduce the results presented in the paper. One thing I am not satisfied with is the lack of description on the noise estimator network \epsilon_\theta. In subsection 4.5, it is described that the network \epsilon_\theta is based on the architecture of ECCT but no details are given in the manuscript. Thus, how to make conditioning with the syndrome weight is not so clear to understand. Of course, a reader can read DDECC.py directly for details but some more descriptions are needed.

[Minor comments]
(1) In many places, SNR is described in EbN0.  It should be Eb/N0 because it is not a product but a ratio.
(2) The work uses the Hamming weight of syndrome as a conditioning parameter (indicating the noise level). The justification of this is heuristically explained based on Fig.3. Are there any convincing ways for this?
(3) The line search procedure improves the overall performance but it produces additional computational overhead. The overhead should be explained.


**Strength And Weaknesses:**

[Strength]
I am working in the field of error correcting codes, especially for LDPC codes. As far as I know, the idea to use the DDPM as a decoding algorithm is definitely novel one.  The idea of the proposed method is natural and reasonable. The method proposed in the paper heavily depends on the idea and the DDPM architecture presented in Ho et al. but  the paper contains non-trivial contributions such as conditioning based on the Hamming weight of the syndrome and line search procedure. Furthermore, the decoding principle presented in the paper has not been discussed in the coding community. The decoding principle presented in the paper  can be a new direction for the further research in coding theory. The experimental results are promising and supports this optimistic view.

[Weekness]
(1) I felt that some details of the algorithm is not well documented in the paper (I will describe it in the next section).
(2) Computational complexity and scalability for longer codes: I got rough understanding on a computational procedure of the decoding process. In order for the proposed method to be a real competitor of the known decoding algorithm such as BP, fair comparisons of the computational complexity will be needed.  For example, BP can be practically applied to LDPC codes of length up to 10^3-10^4. It is better to include some discussions regarding scalability. If this paper concentrates on the ``proof of the concept'', the computational complexity issue is not a large problem so far but some more discussions help the readers.
(3) I would like to see the results of the numerical experiments on AWGN channel in graphs like Figure 4 (SNR versus BER).  It is common to   display decoding performances in such a way in coding community. It makes easier to grasp the difference in decoding performances.

**Summary Of The Paper:**

The paper presents a novel decoding principle for error correcting codes (ECC) based on diffusion models.  The denoising diffusion probability model (DDPM) by Ho et al. can be seen as a basis of the proposed algorithm. A reverse diffusion process in DDPM is used as a decoding process of an ECC. The noise estimator neural network is conditioned on the number of parity errors that  indicates the level of corruption of the original codeword. A line search procedure is introduced for determining a proper diffusion step size. Their experimental results indicates that the proposed algorithm surpasses known state-of-the-art decoding algorithms such as belief propagation(BP). Especially, proposed algorithm has strong advantage for short length codes.

**Summary Of The Review:**

In summary, the decoding principle presented in the paper may indicate a new direction of decoding algorithms. Although I am not so sure whether the proposed algorithm is really competitive against known algorithms such as BP in a practical situation, I am inclining to be positive about this paper because it definitely contains something new from the view point of coding theory.

---

> ### Author Response · Authors · 2022-11-16
> **Authors response**
>
> Thank you for the supportive and comprehensive review.
>
> ## Lack of description of the noise estimator network
>
> Details about the syndrome conditioning in the ECCT model are given in the first paragraph of Section 4.5.  As suggested by the reviewer and in order to make our manuscript self-contained, we now provide an extended description of the ECCT coupled with the parity-checks conditioning in Appendix F.
>
> ## The computational complexity of the method
>
> The proposed framework does not assume a predefined neural decoder. We chose ECCT since it is the current SOTA neural decoder. An extensive analysis of the complexity of ECCT is given in section 6.3 and Appendix G of the ECCT paper https://openreview.net/pdf?id=4F0Pd2Wjl0
>
> Many neural network acceleration methods can be applied in order to greatly reduce the computational complexity of ECCT and thus of DDECC. Finally, BP or its neural variants (e.g. ARBP) cannot ever reach, even at convergence, the performance of our diffusion method even when our method employs extremely low capacity networks, e.g., N=2,d=32.
>
> The method can be applied to arbitrary code lengths. In other domains, Transformers are often run on 512-1024 tokens, which supports the viability of our method for larger codes. However, it may require heavy computational resources or would require implementation on dedicated hardware that is capable of taking the sparsity of the code into account in the self-attention module.
>
> We refer the reader to the citation in the revision. We also added explanations related to the DDECCT complexity in Appendix F2.
>
> ## Graph-like experiments
>
> We followed the performance presentation of previous works (e.g. [1,2]) in a table form, allowing us to display a greater number of codes in a limited space. Following the review, we provide results in the requested format for several codes in Appendix I.
>
> [1] Hyper-graph-network decoders for block codes, *by Nachmani and Wolf.*
>
> [2]  perm2vec: Graph permutation selection for decoding of error correction codes using self-attention, *by Raviv et al.*
>
> ## SNR is described in EbN0
>
> We thank the reviewer for this correction. It was not intended to be understood as a product.
> This is corrected in the revised version.
>
>
> ## Justification of the syndrome conditioning
>
> The denoising process, like many other optimization processes, should employ bigger steps when the current result is far from the solution, and smaller ones when in the vicinity of the optimum. As indicated in Fig. 3, the syndrome is an approximate indication of the proximity to the solution, where near noise-free words the syndrome is close to zero (i.e. very few to no parity-check errors). We emphasized this logic in Section 4.3 of the revised version.
>
>
> ## The line-search procedure overhead
>
> The computation of the syndrome consists of an efficient series of binary operations (xor) inducing a computational complexity that is proportional to the density of the code.
> The line search consists of the parallel computation of the syndrome, over the multiple words obtained for different $\lambda$ values sampled over a predefined grid.
> Thus, the time complexity of the line-search procedure can be reduced to the very efficient computation of the syndrome which can be assumed as constant. Alternatively, without parallelization, the complexity is linear with the grid size.
>
> Following the review, we have elaborated on this in Appendix C of the revised version.

---

> > ### Comment · Reviewer_8AQR · 2022-11-24
> > **Thank you for the revision.**
> >
> > Thank you for your revision and explanation. I increased the recommendation point to 8.

---

### Author Response · Authors · 2022-11-16
**Summary of changes**

We have uploaded a revised version of our manuscript, which contains the recommended corrections, clarifications, and additional results specifically requested by the reviewers.

Following a request by 8AQR, we provide an extended description of the DDECCT in Appendix F1.

Following a request by 8AQR, we added explanations related to the DDECCT complexity in Appendix F2.

Following a request by both 8AQR and U25g, we provide results in the requested graph format for several codes in Appendix I.

Following a request by 8AQR, we have elaborated on the computational complexity of the line-search procedure in Appendix C.

Following a request by YkMp, we added explanations regarding the diffusion process scheduling to Appendix G of the revised manuscript.

Following a request by U25g, we have added in appendix H performance comparisons with the SCL decoder for Polar codes.

Following a request by U25g, we have added in appendix F a description of the DDECCT computational overhead.

We would be happy to continue and improve the manuscript as suggested by the reviewers and the OpenReview community.

---

### Decision · Program_Chairs · 2023-01-20

**Decision:**

Accept: notable-top-25%

**Justification For Why Not Higher Score:**

Although this paper proposes quite original and interesting ideas, they fall in the research area of coding theory, so that I am not really sure about whether people attending ICLR who are not coding theorists would be interested in those ideas.

**Justification For Why Not Lower Score:**

All the reviewers rated this paper highly, mainly due to the novelty of the ideas presented, as well as superior performance of the proposal. I also evaluate this paper to be a good one, well above the acceptance threshold.

**Metareview: Summary, Strengths And Weaknesses:**

This paper proposes a new decoding paradigm for error-correcting codes using denoising diffusion models. The idea of using diffusion models for decoding linear codes is quite original. The numerical results presented in this paper demonstrate superior performance of the proposal over existing decoding methods.

**Note From Pc:**

if the above contains the word "oral" or "spotlight" please see: "oral" presentation means -> notable-top-5% and "spotlight" means -> notable-top-25%. As stated in our emails, we are disassociating presentation type from AC recommendations